# $G^2$-Reader: Dual Evolving Graphs for Multimodal Document QA

Yaxin Du [* 1]  Junru Song [* 1]  Yifan Zhou [* 1]  Cheng Wang [1]  Jiahao Gu [1]  Zimeng Chen [1]  Menglan Chen [1]
Wen Yao [2]  Yang Yang [2]  Ying Wen [1 3]  Siheng Chen [1]

## Abstract

Retrieval-augmented generation is a practical paradigm for question answering over long documents, but it remains brittle for multimodal reading where text, tables, and figures are interleaved across many pages. First, flat chunking breaks document-native structure and cross-modal alignment, yielding semantic fragments that are hard to interpret in isolation. Second, even iterative retrieval can fail in long contexts by looping on partial evidence or drifting into irrelevant sections as noise accumulates, since each step is guided only by the current snippet without a persistent global search state. We introduce $G^2$-Reader, a dual-graph system, to address both issues. It evolves a Content Graph to preserve document-native structure and cross-modal semantics, and maintains a Planning Graph, an agentic directed acyclic graph of sub-questions, to track intermediate findings and guide stepwise navigation for evidence completion. On VisDoMBench across five multimodal domains, $G^2$-Reader with Qwen3-VL-32B-Instruct reaches 66.21% average accuracy, outperforming strong baselines and a standalone GPT-5 (53.08%). Code is available: https://github.com/DorothyDUUU/$G2$_Reader.

## 1. Introduction

Large language models (LLMs) (Zhao et al., 2023) have demonstrated remarkable capabilities in assisting humans with reading, analysis, and reasoning over complex content. However, real-world reading tasks frequently involve long documents or large corpora that exceed the limited context window of LLMs (Beltagy et al., 2020; Xiong et al., 2024). To address this constraint, Retrieval-Augmented Generation (RAG) (Lewis et al., 2020) has emerged as a dominant paradigm, transforming long-context understanding into a retrieve-then-read workflow by grounding generation on externally retrieved evidence (Izacard & Grave, 2021; Guu et al., 2020; Karpukhin et al., 2020).

In practice, documents are often multimodal, interleaving natural language with tables and figures over many pages. A key feature of such documents is that they encode tightly coupled heterogeneous evidence (*e.g.*, text, tables, and figures) that mutually ground each other (Suri et al., 2025; Cho et al., 2024). This inherent complexity poses a fundamental crisis for current RAG systems, manifesting in two coupled dimensions: ❶ **the Representation Challenge:** Despite this coupling, standard RAG pipelines flatten documents into independent chunks, which breaks document-native structure and cross-modal alignment (*e.g.*, separating a figure/table from its caption, referring text, and explanatory paragraph) (Faysse et al., 2024). As a result, retrieved contexts are often semantic fragments that are hard to interpret in isolation. ❷ **the Retrieval Challenge:** Traditional RAG is largely one-shot, which is brittle for long contexts; iterative retrieval (Jiang et al., 2023a; Trivedi et al., 2023a) was introduced to refine queries step by step. In long contexts, iterative retrieval often fails by looping on partial evidence or drifting into irrelevant sections as noise accumulates (Lin et al., 2025; Gong et al., 2025). This happens because each step is guided only by the current snippet, with no persistent global state of the search.

To tackle these two problems, we introduce $G^2$-**Reader**, a dual-**G**raph system for long-document multimodal QA. Its key idea is that RAG should advance on both evidence representation and retrieval-reasoning fronts, and we achieve this through a dual-graph mechanism: ❶ **Content Graph** ($\mathcal{G}_C$)**:** Instead of flat chunking, we ground multimodal evidence in a structured topology that preserves document-native layouts and cross-modal alignments. It enables message passing, allowing evidence representations to evolve contextually and restore global awareness. ❷ **Planning Graph** ($\mathcal{G}_P$)**:** We maintain an agentic directed acyclic graph (DAG) as an explicit structured carrier for the reasoning process. By decomposing complex questions into sub-questions, it iteratively updates the reasoning state with intermediate findings to guide precise navigation over the underlying content.

---

[*]Equal contribution [1]Shanghai Jiao Tong University [2]Intelligent Game and Decision Laboratory [3]Shanghai Innovation Institute. Correspondence to: Siheng Chen <sihengc@sjtu.edu.cn>.

*Proceedings of the $43^{rd}$ International Conference on Machine Learning*, Seoul, South Korea. PMLR 306, 2026. Copyright 2026 by the author(s).

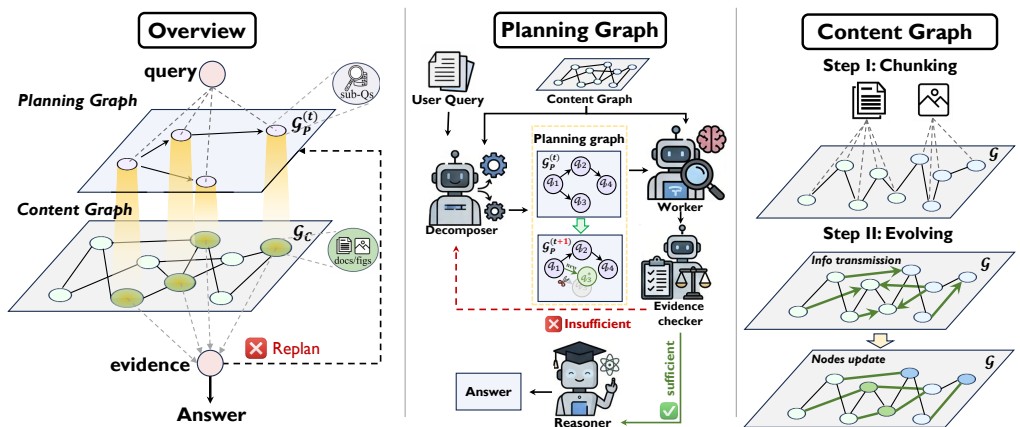

*Figure 1.* Overview of the $G^2$-Reader framework (Left). $G^2$-Reader bridges the gap between offline document indexing and online inference through a dual-graph architecture: (Right) The **Content Graph** ($\mathcal{G}_C$) provides a structured multimodal representation of documents with evolved node attributes; (Middle) The **Planning Graph** ($\mathcal{G}_P$) serves as an explicit, dynamic reasoning state that iteratively orchestrates sub-question decomposition and evidence assembly.

Concretely, $G^2$-Reader first converts each document into a heterogeneous multimodal **content graph** whose nodes are (a) paragraphs and (b) multimodal elements (tables/figures with captions), and whose edges encode document-native relations such as reference and multimodal associations. Through lightweight message passing, each node perceives its broader context, establishing meaningful connections and preventing the semantic fragmentation common in traditional chunking. Secondly, over this stable structural foundation, $G^2$-Reader orchestrates reasoning via the Planning Graph, a DAG of sub-questions. Rather than a blind one-shot lookup, the system performs context-aware exploration: each retrieval step informs and refines the subsequent plan. This iterative feedback loop allows the system to incrementally assemble a coherent body of evidence, moving from initial uncertainty to a complete and logically sound answer. By grounding the dynamic reasoning path within a fixed structural prior, $G^2$-Reader ensures that the retrieved evidence is both structurally intact and logically sufficient.

Empirical evaluations across five multimodal domains (including slides, web pages, academic papers, and textbooks) demonstrate $G^2$-Reader's superiority and robustness. Utilizing the open-source Qwen3-VL-32B-Instruct, our system achieves an average score of 66.21%, significantly surpassing all baselines and even a standalone GPT-5 (53.08%). These results validate that our dual-graph architecture effectively empowers open-source models to outperform powerful closed-source frontiers by providing a structured foundation for complex evidence assembly.

Our main contributions are:

❶ We identify two coupled limitations of current multimodal RAG systems: fragmented evidence representation and unstable long-context retrieval, and therefore propose $G^2$-Reader, a dual-graph system to address both.

❷ We introduce a Content Graph to preserve semantic structures within multimodal documents, and a Planning Graph to maintain reasoning state and guide stepwise navigation for evidence assembly.

❸ On VisDoMBench across five domains, $G^2$-Reader with an open-source backbone achieves the best overall accuracy. Comprehensive ablations further validate the effectiveness and complementarity of each component.

## 2. Related Work

### 2.1. Memory Construction for RAG

Retrieval-augmented generation (RAG) critically depends on how external memory is represented. Early approaches largely treat documents as flattened collections of text chunks, discarding document structure and multimodal context (Lewis et al., 2020). To overcome this limitation, a substantial line of work focuses on structuring document memory prior to retrieval, organizing content into graphs over entities, passages, or topics in order to encode relations such as co-reference, topical continuity, and cross-section dependency (Edge et al., 2024; Li et al., 2025; Chen et al., 2025; Hu et al., 2025; Shen et al., 2025; Liao, 2025; Xu et al., 2025; Chhikara et al., 2025). In parallel, multimodal RAG systems extend structured memory to visually rich documents by retrieving or indexing over images, layouts, tables, and figures, or by aligning heterogeneous elements within multimodal graphs or unified indices (Yu et al., 2024a; Faysse et al., 2024; Wan & Yu, 2025; Guo et al., 2025). Despite the progress, memory construction in existing systems is typically treated as a single-pass preprocessing step, relying on heuristic extraction or standalone summariza-

tion, and retrieval predominantly matches queries against embeddings of raw textual or visual content. In contrast, our method formulates memory construction as an *iterative, VLM-driven consolidation process*, where multimodal memory units are evolved based on their neighborhood to distill core semantics and uncover document-intrinsic relations, and tightly coupled into a memory graph for more precise and comprehensive retrieval beyond surface-level similarity.

## 2.2. Agentic Retrieval with Reasoning

An orthogonal line of research improves RAG by making retrieval adaptive to intermediate reasoning. Early approaches expose reasoning steps to guide information seeking, such as explicit sub-question decomposition (Press et al., 2023) or interleaving reasoning with retrieval or tool use (Yao et al., 2022; Trivedi et al., 2023b). Subsequent methods refine retrieval control through uncertainty-based triggers, self-reflection signals, or autonomous multi-turn interaction with retrievers (Jiang et al., 2023b; Asai et al., 2023; Yu et al., 2024b; Shao et al., 2023; Hui et al., 2025). System-level extensions distribute retrieval or orchestration across multiple agents to scale context size or pipeline complexity (Nguyen et al., 2025; Liu et al., 2025; Zhang et al., 2025). Across these methods, reasoning is typically modeled as a *linear sequence* of steps or actions, and retrieved evidence is accumulated as transient context without explicit representation of dependencies or coverage. Our approach departs from this paradigm by formulating information seeking as an explicit *directed acyclic graph*, where the retrieval process is itself planned and revised. Evidence is attached to graph nodes, evaluated for sufficiency, and used to trigger structural modification. Crucially, this structured reasoning process operates over the semantically consolidated multimodal memory described above, enabling systematic coordination between how knowledge is represented and how it is planned, verified, and composed during inference.

# 3. Methodology: $G^2$-Reader

$G^2$-Reader is built upon a dual-graph paradigm that tightly couples two core components: (i) a multimodal **Content Graph** providing structured document representation, and (ii) a **Planning Graph** explicitly anchoring the reasoning process to interdependent decisions. We first formalize the problem of RAG (§3.1), then present the construction of the Content Graph (§3.2). This is followed by the iterative refinement of the Planning Graph for evidence-driven reasoning and answer synthesis (§3.3).

## 3.1. Problem Formulation

Given a query $Q$ and a document corpus $\mathcal{D}$, retrieval-augmented generation aims to generate an answer $A$ grounded in external evidence. The generation process can

be viewed as conditioning on a set of supporting evidence $E \subseteq \mathcal{D}$, yielding the posterior: $A^* = \arg\max_A P_\theta(A \mid Q, E)$, where $P_\theta$ denotes the conditional likelihood modeled by a parametric generator (*e.g.*, an LLM or VLM). The quality of the generated answer therefore critically depends on the construction of an appropriate evidence set.

We define the core objective of RAG as identifying a *minimally sufficient* evidence set $E^*$ that satisfies the information requirements implied by $Q$:

$$E^* = \arg\min_E |E| \quad \text{s.t.} \quad E \models Q,$$

where $E \models Q$ indicates that $E$ provides adequate support to answer $Q$ under its implicit logical and semantic constraints. In practice, directly optimizing $E^*$ over the raw corpus $\mathcal{D}$ is intractable, especially for long, multimodal documents.

$G^2$-Reader addresses this challenge by factorizing evidence construction into two structured spaces. The document corpus $\mathcal{D}$ is first converted into a static multimodal *Content Graph* $\mathcal{G}_C$ that encodes document elements and their relations. An agent then incrementally builds a *Planning Graph* $\mathcal{G}_P$, modeled as a directed acyclic graph whose nodes represent intermediate sub-questions.

Under this formulation, approximating $E^*$ reduces to navigating $\mathcal{G}_P$ over $\mathcal{G}_C$ and retrieving, for each planning node $q_i$, a structurally grounded evidence subgraph $G_{q_i} \subseteq \mathcal{G}_C$:

$$E^* \approx \bigcup_{q_i \in \text{Nodes}(\mathcal{G}_P)} G_{q_i}.$$

$\mathcal{G}_P$ is iteratively refined based on evidence sufficiency, while $\mathcal{G}_C$ remains fixed during inference. This separation decouples *how evidence is represented* from *how reasoning is structured*, providing a principled foundation for agentic, coverage-aware retrieval and reasoning.

## 3.2. Content Graph

To preserve document-native structure and cross-modal semantics that are often lost in flat RAG architectures, we represent a multi-document corpus as a heterogeneous multimodal *Content Graph* $\mathcal{G}_C = (V, \mathcal{E})$. Each node in $\mathcal{G}_C$ corresponds to an atomic information unit grounded in the original document, while edges encode both structural proximity and semantic relations. This representation ensures that multimodal evidence remains explicitly anchored to its source context and can be retrieved and composed in a structure-aware manner.

### 3.2.1. AUTOMATED GRAPH CONSTRUCTION

**Multimodal Parsing and Node Generation.** Each document is first processed by a multimodal parser (*e.g.,* Mineru (Wang et al., 2024) or DeepSeek-OCR (Wei et al., 2025)) that extracts, while retaining layout and reading order. These elements are segmented into atomic units, forming the

node set $V$. Each node $v_i \in V$ is represented as a tuple $(c_i, attr_i, h_i)$, where $c_i$ denotes the raw multimodal content and $h_i \in \mathbb{R}^d$ is its latent embedding. To align heterogeneous modalities into a unified semantic space, we associate each node with an attribute set $attr_i = (s_i, k_i)$, where $s_i$ is a dense summary and $k_i$ is a set of discriminative keywords, both generated by a VLM. The latent representation $h_i$ is calculated as $h_i = \text{Encoder}_\phi(s_i \oplus k_i)$, where $\text{Encoder}_\phi$ is a pretrained embedding model, and $\oplus$ denotes string concatenation. By grounding node embeddings in semantically distilled summaries rather than raw content, this formulation enables consistent representation across modalities and supports efficient retrieval. These initial representations are denoted as $s_i^{(0)}, k_i^{(0)}$, and $h_i^{(0)}$ and serve as the starting point for the iterative evolution described below.

**Edge Initialization.** We initialize the Content Graph topology by linking each node $v_i$ to a sliding window of its neighbors in the reading order: $\mathcal{E}^{(0)} = \{(v_i, v_j) \mid 1 \leq |i - j| \leq w\}$. This lightweight initialization preserves local coherence and provides a reliable scaffold for subsequent refinement.

### 3.2.2. JOINT GRAPH EVOLUTION OF TOPOLOGY AND NODE ATTRIBUTES

Although each node is initially equipped with local summaries and keywords and connected via adjacency-based links, such representations remain limited: node attributes are often context-deficient and weakly discriminative, while layout-induced links capture only surface proximity and fail to reflect non-local semantic dependencies. To overcome these limitations, $G^2$-Reader performs an iterative *joint evolution* of node attributes and graph topology.

**Candidate Neighborhood Construction.** At evolution step $t$ ($t = 0, \cdots, T - 1$), for each node $v_i$, we construct a candidate neighborhood $\mathcal{C}_i^{(t)}$ that combines semantic proximity and the currently evolved topological structures:

$$\mathcal{C}_i^{(t)} = \underset{j \in V \setminus \{i\}}{\text{TopK}} \left( \text{Sim}(h_i^{(t)}, h_j^{(t)}) \right) \cup \mathcal{N}^{(t)}(v_i),$$

where "Sim" denotes similarity measures in the latent space, such as cosine similarity, and $\mathcal{N}^{(t)}(v_i)$ is the neighborhood of $v_i$ indicated by $\mathcal{E}^{(t)}$. This design ensures that each node is exposed both to potentially relevant but unlinked nodes and to its current relational context.

**VLM-based Joint Update.** Given the node content $c_i$, its current attributes $(s_i^{(t)}, k_i^{(t)})$, and the candidate set $\mathcal{C}_i^{(t)}$, a vision-language model produces a joint update that refines both representation and connectivity:

$$(s_i^{(t+1)}, k_i^{(t+1)}, \mathcal{N}^{(t+1)}(v_i))$$
$$\leftarrow \text{VLM}\left(c_i, s_i^{(t)}, k_i^{(t)}, \{(c_j, s_j^{(t)}, k_j^{(t)})\}_{v_j \in \mathcal{C}_i^{(t)}}\right).$$

Concretely, the VLM updates node attributes by (i) enrich-ing the summary with missing contextual information inferred from related nodes, (ii) resolving implicit references and dependencies, and (iii) emphasizing aspects that distinguish the node from semantically adjacent content. The resulting attributes therefore provide a more faithful and discriminative abstraction of each node's role within the document, which is critical for accurate retrieval. Simultaneously, the VLM identifies a sparse set of *meaningful* links, $\mathcal{N}^{(t+1)}(v_i))$, based on explicit logical relations, such as explanation, elaboration, comparison, causality, or cross-modal grounding. These links are not induced by similarity alone, but reflect substantive dependencies that support multi-step reasoning.

**Parallel Graph Evolution.** Since the evolution operator is inherently local to each node, in practice, node updates can be executed asynchronously, such that once a node finishes its update, its refined attributes can be immediately consumed by other nodes still undergoing evolution. This asynchronous implementation preserves the semantics of local joint refinement while substantially improving efficiency, especially for long documents. After $T$ iterations, the evolved Content Graph $\mathcal{G}_C^{(T)}$ is fixed and serves as the static evidence space during subsequent inference.

### 3.2.3. STRUCTURED SUBGRAPH READOUT

Evidence is retrieved from $\mathcal{G}_C$ via a structure-aware readout operator. Given a query $q$, we compute its embedding $h_q = \text{Encoder}_\phi(q)$ and ranks all nodes by cosine similarity $\text{Sim}(v_i, q) = \frac{h_i^\top h_q}{\|h_i\| \|h_q\|}$, yielding an ordered list $V_{\text{ranked}}$. We then construct the output node set $V_{\text{out}}$, initialized as $\varnothing$, by greedily selecting nodes from $V_{\text{ranked}}$ and expanding each selection with its immediate neighbors $\mathcal{N}(v)$ in $\mathcal{G}_C$:

$$V_{\text{out}} \leftarrow V_{\text{out}} \cup (\{v\} \cup \mathcal{N}(v)).$$

The process terminates once $|V_{\text{out}}| \geq k$, where $k$ denotes a node budget. The induced subgraph $G_q$ provides compact, context-aware evidence for downstream reasoning.

### 3.3. Dynamic Planning Graph Evolution

Given the evolved Content Graph $\mathcal{G}_C^{(T)}$, $G^2$-Reader constructs an explicit planning Graph $\mathcal{G}_P$ to model the reasoning process required to answer a query. Interaction with $\mathcal{G}_C^{(T)}$ is mediated by the structured subgraph readout (§ 3.2.3) operator. The Planning Graph itself is dynamically refined across iterations in response to evidence sufficiency. The whole process, as illustrated in Figure 1, is realized through a set of distinct yet coordinated roles, including a `decomposer`, a `worker`, an `evidence checker`, and a `reasoner`, all instantiated by a VLM with role-specific prompts.

### 3.3.1. INITIAL PLANNING GRAPH CONSTRUCTION

Conditioned on the input query $Q$, $G^2$-Reader first performs a lightweight probing retrieval to obtain a coarse overview of the evidence landscape, based on which the `decomposer` generates an initial Planning Graph: $\mathcal{G}_P^{(0)} = (\mathcal{Q}, \Delta)$, where each node $q_i \in \mathcal{Q}$ corresponds to an atomic sub-question, and directed edges $\Delta$ encode dependency relations specifying partial ordering and logical prerequisites among reasoning steps. By construction, $\mathcal{G}_P^{(0)}$ forms a directed acyclic graph (DAG), ensuring a well-defined execution order and preventing cyclic reasoning.

### 3.3.2. PLANNING GRAPH EXECUTION AND REFINEMENT

**Topological Execution.** Since $\mathcal{G}_P$ is a directed acyclic graph, $G^2$-Reader derives a valid execution order $\pi = (q_{\pi_1}, q_{\pi_2}, \ldots, q_{\pi_n})$ using Kahn's algorithm, such that for any directed edge $(q_i, q_j) \in \Delta$, node $q_i$ precedes $q_j$ in $\pi$. Planning nodes are then executed following this order, ensuring that the reasoning process respects the partial order specified by the Planning Graph.

**Local Evidence Retrieval and Reasoning.** For each sub-question $q_i$, $G^2$-Reader invokes the structured subgraph readout operator to retrieve a local evidence subgraph $G_{q_i} \subseteq \mathcal{G}_C^{(T)}$. Node-level reasoning is then performed by conditioning the `worker` on both the retrieved evidence and the current sub-question. In addition, when available, intermediate answers produced by child nodes in the Planning Graph are incorporated into the reasoning context as supplementary grounded information. Formally, the intermediate answer for node $q_i$ is generated as:

$$a_i = \text{VLM}_{\text{worker}} \left( q_i, \; G_{q_i} \oplus \bigoplus_{q_c \in \text{Child}(q_i)} a_c \right),$$

where $\text{Child}(q_i)$ denotes the set of child nodes of $q_i$, and $a_c$ is the intermediate answer generated for child node $q_c$. This design allows higher-level planning nodes to leverage validated conclusions obtained at lower-level nodes, while keeping evidence retrieval itself local and self-contained.

**Evidence Aggregation and Sufficiency Check.** After all planning nodes have been executed, $G^2$-Reader aggregates the intermediate answers $\{(q_i, a_i)\}$ and evaluates whether the accumulated evidence is sufficient to answer the original query $Q$. This global verification step is performed by the `evidence checker` as follows:

$$(\texttt{sufficient}, \mathcal{G}_{\text{gaps}}) = \text{VLM}_{\text{checker}} \left( Q, \; \{(q_i, a_i)\}_{q_i \in \mathcal{G}_P^{(\tau)}} \right),$$

where `sufficient` indicates whether the current evidence satisfies the information requirements of $Q$, and $\mathcal{G}_{\text{gaps}}$ identifies missing, ambiguous, or under-explored aspects when the criterion is not met.

**Iterative Planning Graph Refinement.** If the evidence suf-

ficiency check fails, $G^2$-Reader refines the Planning Graph by generating an updated decomposition that explicitly targets the identified gaps:

$$\mathcal{G}_P^{(\tau+1)} = \text{VLM}_{\text{decomposer}} \left( Q, \; \{(q_i, a_i)\}, \; \mathcal{G}_{\text{gaps}}, \; \mathcal{G}_P^{(\tau)} \right).$$

The refined graph may introduce new sub-questions, modify dependency relations, or remove redundant nodes. This execution–verification–refinement loop continues until the evidence sufficiency criterion is satisfied or a maximum number of refinement iterations $\tau_{\max}$ is reached.

### 3.3.3. EVIDENCE-GROUNDED ANSWER SYNTHESIS

Upon termination, $G^2$-Reader converges to a final Planning Graph $\mathcal{G}_P^*$ and an associated set of verified evidence:

$$E^* = \{(q_i, G_{q_i}) \mid q_i \in \text{Nodes}(\mathcal{G}_P^*)\}.$$

The final answer is generated by conditioning `reasoner` on both the original query and structured evidence:

$$A = \text{VLM}_{\text{reasoner}}(Q, E^*).$$

By decoupling structured evidence representation (Content Graph), evidence access (structured subgraph readout), and reasoning control (Planning Graph), $G^2$-Reader ensures that generation is grounded in a logically sufficient yet minimally redundant evidence set, substantially reducing hallucinations caused by isolated or distractor fragments.

## 4. Experiments

We evaluate $G^2$-Reader on VisDoMBench, a multimodal, multi-document QA benchmark. Experiments assess end-to-end effectiveness of $G^2$-Reader against competitive baselines (§4.2), and analyze how its key components contribute to performance (§4.3) and interpretability (§4.5).

### 4.1. Experimental Setup

**Dataset.** We conduct our experiments on VisDoMBench (Suri et al., 2025), a multimodal QA benchmark designed for challenging multi-document grounded reasoning. For fair comparison and computational efficiency, we adopt a unified setting where each question is paired with five documents containing both relevant evidence and distractors. VisDoMBench contains 2,271 samples spanning five subsets that differ in document type and evidence modality: (i) **FetaTab** (Hui et al., 2024; Nan et al., 2022), table-centric factual QA over Wikipedia pages; (ii) **PaperTab** (Dasigi et al., 2021), table-centric QA over scientific papers; (iii) **SPIQA** (Pramanick et al., 2024), cross-modal scientific paper QA involving figures, tables and text; (iv) **SciGraphQA** (Li & Tajbakhsh, 2023), a synthetic multi-turn VQA dataset centered on academic graphs; (v) **SlideVQA** (Tanaka et al., 2023), containing multi-page slide decks that requires cross-slide, multi-hop, and numerical reasoning.

*Table 1.* Main results on the full VisDoMBench benchmark. All results are averaged over three runs, with "±" denoting standard deviation. $G^2$-Reader (66.21%) overall outperforms representative single-VLM and RAG baselines across five multimodal domains.

| | SPIQA | FetaTab | PaperTab | SciGraphQA | SlideVQA | Average |
|---|---|---|---|---|---|---|
| | | | Single-VLM | | | |
| **GPT-5** | 55.22 ± 0.09 | 63.94 ± 0.31 | 37.08 ± 0.12 | 64.08 ± 0.32 | 45.06 ± 0.10 | 53.08 ± 0.11 |
| **Qwen3-VL-32B** | 29.86 ± 0.08 | 37.39 ± 0.36 | 34.32 ± 0.27 | 23.06 ± 0.22 | 24.87 ± 0.24 | 29.90 ± 0.11 |
| | | | RAG Baseline (Qwen3-VL-32B-Instruct) | | | |
| **Deepseek-OCR** | 63.60 ± 0.40 | 70.32 ± 0.12 | 51.58 ± 0.24 | 61.91 ± 0.40 | 65.69 ± 0.12 | 62.62 ± 0.12 |
| **GraphRAG** | 62.65 ± 0.20 | 61.35 ± 0.19 | 42.90 ± 0.00 | 65.76 ± 0.38 | 21.68 ± 0.00 | 50.86 ± 0.09 |
| **LightRAG** | 73.88 ± 0.00 | 64.71 ± 0.38 | 51.02 ± 0.04 | 75.00 ± 0.01 | 29.63 ± 0.01 | 63.74 ± 0.13 |
| **MMGraphRAG** | 69.91 ± 0.23 | 72.40 ± 0.55 | 56.36 ± 0.58 | 64.11 ± 0.25 | 54.20 ± 0.15 | 63.40 ± 0.16 |
| **VisDoMRAG** | 75.44 ± 0.00 | 61.02 ± 0.50 | 56.21 ± 0.15 | 63.36 ± 0.14 | 69.03 ± 0.36 | 65.01 ± 0.13 |
| **RAGAnything** | 67.69 ± 0.96 | 57.76 ± 0.24 | 42.02 ± 1.35 | 41.60 ± 2.60 | 52.18 ± 0.49 | 52.25 ± 0.63 |
| **ViDoRAG** | 68.18 ± 0.46 | 58.74 ± 0.38 | 43.67 ± 0.15 | 37.86 ± 0.14 | 71.71 ± 0.11 | 56.03 ± 0.13 |
| **MA-RAG** | 45.52 ± 0.22 | 27.70 ± 0.19 | 33.43 ± 0.45 | 29.32 ± 0.25 | 29.40 ± 0.21 | 33.07 ± 0.11 |
| $G^2$-**Reader** | 73.19 ± 0.21 | 66.89 ± 0.11 | 57.10 ± 0.21 | 61.56 ± 0.11 | 72.31 ± 0.00 | **66.21 ± 0.08** |

**Baselines.** We compare $G^2$-Reader against four categories of baselines: **(i) Single VLMs**, including a proprietary frontier model (GPT-5) and a strong open-source model (Qwen3-VL-32B-Instruct), which are directly prompted with raw PDF pages to generate answers; **(ii) Simple RAG**, a conventional pipeline that applies DeepSeek-OCR to extract text from document pages, followed by vector-based retrieval and answer generation; **(iii) Structured text-centric RAG**, a strong graph-based retrieval baseline operating primarily over textual content (GraphRAG Edge et al., 2024, LightRAG (Guo et al., 2024)), and **(iv) Advanced Multimodal RAG**, representative state-of-the-art systems designed for visually rich documents, including MMGraphRAG (Wan & Yu, 2025), VisDoMRAG (Suri et al., 2025), RAGAnything (Guo et al., 2025), ViDoRAG (Wang et al., 2025), and MA-RAG (Nguyen et al., 2025). Further implementation details are provided in Appendix B.

**Evaluation Metrics.** We measure end-to-end **Accuracy** assessed by an LLM-based evaluator (GPT-4), which compares the generated response against the ground-truth answer. This evaluation protocol reflects the system's overall ability to understand the query, retrieve relevant evidence, and synthesize an accurate response, while being robust to lexical and stylistic variations. See Appendix A for details.

**Implementation.** Key hyperparameters are set as follows: the Content Graph is evolved for $T = 3$ iterations, and the Planning Graph is refined for up to $\tau_{max} = 3$ iterations. For all retrieval-based approaches, the number of retrieved candidates per query is fixed to $k = 5$ to ensure a consistent retrieval budget. MinerU (Wang et al., 2024) is employed for multimodal document parsing. All reported results are averaged over three independent runs. And ablation study are using randomly sampled 250 samples from 5 subsets for ablation studies.

## 4.2. Main Results

To assess end-to-end effectiveness under a controlled retrieval budget, we compare single-VLM baselines, representative RAG baselines, and $G^2$-Reader using a unified node budget ($k=5$). Table 1 reports accuracy across the five VisDoMBench subsets.

**Comparison with Single VLMs.** As demonstrated in Table 1, $G^2$-Reader significantly outperforms both open-source and proprietary single-VLM baselines. While GPT-5 achieves an average accuracy of 53.08%, $G^2$-Reader (66.21%)—powered by the open-source Qwen3-VL-32B—surpasses it by 13.13 absolute percentage points. More notably, pure zero-shot inference with Qwen3-VL-32B (29.90%) struggles severely with the long-context and multi-document nature of VisDoMBench. $G^2$-Reader achieves a **121% relative improvement** over its base model. This suggests that for complex multimodal reasoning, sophisticated architecture for evidence organization and iterative planning can be more effective than raw scaling or proprietary training.

**Comparison with Other RAG Systems.** Overall, $G^2$-Reader achieves the best average performance (66.21%), outperforming representative multimodal RAG systems such as VisDoMRAG (65.01%) and specialized graph-based systems like GraphRAG (50.86%). The improvements are most pronounced on **FetaTab** (66.89%) and **SlideVQA** (72.31%). We attribute this to the *alignment* between our Content Graph and Planning Graph: while standard RAG systems (*e.g.*, Deepseek-OCR + RAG, 65.69%) retrieve clusters of chunks that may lack logical flow, $G^2$-Reader's dual-graph architecture allows it to navigate across fragmented pages or table sections while maintaining a persistent reasoning state. On datasets like **SPIQA** and **SciGraphQA**, while our system is superior at *global evidence assembly*, it remains competitive but slightly trails specialized base-

*Table 2.* Comparison of retrieval and reasoning strategies. Multimodal modeling (MM), the Content Graph ($\mathcal{G}_C$), and the Planning Graph ($\mathcal{G}_P$) each contribute to performance, with their combination achieving the best results. "FT", "PT", and "SCGQA" denote FetaTab, PaperTab, and SciGraphQA, respectively, applied to Table 3 and Table 4.

| Config | | | SPIQA | FT | PT | SCGQA | SlideVQA | Avg |
|---|---|---|---|---|---|---|---|---|
| MM | $\mathcal{G}_c$ | $\mathcal{G}_p$ | | | | | | |
| ✗ | ✗ | ✗ | 62.0 | 68.0 | 54.0 | 50.0 | 40.0 | 54.8 |
| ✓ | ✗ | ✗ | 72.0 | 64.0 | 60.0 | 54.0 | 66.0 | 63.2 |
| ✓ | ✗ | ✓ | 76.0 | 68.0 | 54.0 | 54.0 | 70.0 | 64.4 |
| ✓ | ✓ | ✗ | 74.0 | 64.0 | 58.0 | 62.0 | 60.0 | 63.6 |
| ✓ | ✓ | ✓ | 80.0 | 66.0 | 64.0 | 56.0 | 74.0 | **68.0** |

*Table 3.* Effect of iterative Planning Graph refinement. Performance improves with additional refinement rounds and saturates after three iterations.

| $\tau_{\max}$ | SPIQA | FT | PT | SCGQA | SlideVQA | Avg |
|---|---|---|---|---|---|---|
| 0 | 76.0 | 70.0 | 52.0 | 60.0 | 66.0 | 64.8 |
| 1 | 78.0 | 70.0 | 58.0 | 55.0 | 62.0 | 64.6 |
| 2 | 82.0 | 74.0 | 52.0 | 56.0 | 64.0 | 65.6 |
| 3 | 80.0 | 66.0 | 64.0 | 56.0 | 74.0 | **68.0** |
| 4 | 80.0 | 68.0 | 54.0 | 54.0 | 66.0 | 64.4 |
| 5 | 76.0 | 64.0 | 64.0 | 56.0 | 68.0 | 65.6 |

lines (like VisDoMRAG on SPIQA) due to the resolution constraints of initial graph nodes when answering questions about extremely fine-grained details in complex figures.

## 4.3. Ablation Studies

### 4.3.1. COMPONENT ANALYSIS

Table 2 presents a factorized ablation of $G^2$-Reader, where each row corresponds to a configuration. *Multimodal* indicates whether visual elements are used in retrieval and generation; *Content Graph* denotes whether evidence is retrieved from the evolved Content Graph (vs. a flattened baseline without abstraction or links); and *Planning Graph* denotes whether inference uses an explicit DAG-based planning process (vs. single-shot retrieval and direct answering). The results show three clear trends: ❶ **Multimodal evidence is necessary.** Adding visual evidence improves the average accuracy from 54.8% to 63.2% (+8.4%), with the largest gain on SlideVQA (40.0%→66.0%, +26.0%). ❷ **Both components are individually beneficial.** On top of the vanilla multimodal baseline, enabling the Planning Graph increases performance to 64.4 (+1.2%), while enabling the Content Graph increases it to 63.6 (+0.4%). ❸ **The two graphs are complementary.** Combining both components yields the best overall accuracy, improving over the vanilla multimodal baseline by +4.8%; this gain exceeds the sum of the individual gains (+1.2% and +0.4%), indicating synergy between structured evidence (Content Graph) and focused decomposition (Planning Graph).

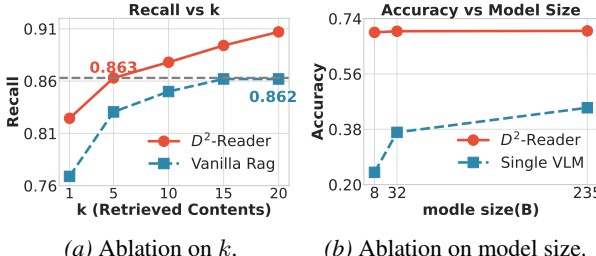

*(a) Ablation on $k$.*      *(b) Ablation on model size.*

*Figure 2.* Analysis of $G^2$-Reader Characteristics. (a) Comparison between $G^2$-Reader and Vanilla RAG over various numbers of retrieved content $k$. (b) Comparison of $G^2$-Reader and Single VLM over various model sizes.

### 4.3.2. RETRIEVAL-SIDE ABLATION: PLANNING GRAPH

Table 3 varies the maximum number of Planning Graph (DAG) refinement rounds $\tau_{\max}$. The results show two clear trends: ❶ **Moderate refinement is beneficial.** Moving from no refinement ($\tau_{\max}$=0) to the best setting ($\tau_{\max}$=3) improves average accuracy from 64.8% to 68.0%, with the largest gain on SlideVQA (+8.0%) and a notable improvement on PaperTab (+12.0%). ❷ **Over-refinement hurts.** Increasing $\tau_{\max}$ beyond 3 reduces performance (e.g., 68.0%→64.4% at $\tau_{\max}$=4), suggesting that extra replanning may introduce redundant or weakly grounded subquestions and lead to noisier evidence aggregation under a fixed budget.

### 4.3.3. REPRESENTATION-SIDE ABLATIONS: CONTENT GRAPH

Table 4 examines the impact of Content Graph evolution ($T$) and structured subgraph readout (SubG.). The results show three trends. ❶ **Evolution helps up to a point.** Increasing evolution from $T$=0 to $T$=3 improves average accuracy by +6.2%, indicating that iterative refinement yields more context-aware node representations. Performance then drops at larger $T$ (e.g., 63.6% at $T$=5), consistent with an over-smoothing effect from repeated propagation (Cai & Wang, 2020). ❷ **A lightweight variant is competitive.** The rule-based Lite variant at $T$=3* reaches 67.0, close to full VLM-based evolution, offering a practical efficiency–accuracy trade-off (Appendix C). ❸ **Neighbor-aware readout improves evidence completion.** Disabling structured subgraph readout at $T$=3 reduces the average from 68.0% to 66.0%, with the largest drop on SlideVQA (74.0%→62.0%), suggesting that neighbor-aware retrieval is particularly helpful when evidence is fragmented across pages.

## 4.4. Analysis

Figure 2 analyzes the robustness of $G^2$-Reader regarding retrieval budgets and model scaling. As shown in Figure 2a, $G^2$-Reader's recall improves steadily with $k$, consistently

*Table 4.* Effect of Content Graph evolution and structured subgraph readout. Performance improves with iterative evolution up to three rounds, while neighbor-aware retrieval is particularly beneficial for scenarios with fragmented evidence. "$\star$" denotes a lite variant of $G^2$-Reader, where VLM-based evolution is replaced by rule-based updating. **Bold** and underlined numbers indicate the best and second-best results, respectively.

| $\mathcal{G}_C$ Config | | SPIQA | FT | PT | SCGQA | SlideVQA | Avg |
|---|---|---|---|---|---|---|---|
| $T$ | SubG. | | | | | | |
| 0 | ✓ | 70.0 | 64.0 | 56.0 | 48.0 | 71.0 | 61.8 |
| 1 | ✓ | 74.0 | 72.0 | 56.0 | 52.0 | 66.0 | 64.0 |
| 2 | ✓ | 76.0 | 72.0 | 58.0 | 60.0 | 62.0 | 65.6 |
| 3 | ✓ | 80.0 | 66.0 | 64.0 | 56.0 | 74.0 | **68.0** |
| 3$\star$ | ✓ | 85.0 | 68.0 | 66.0 | 54.0 | 62.0 | 67.0 |
| 3 | ✗ | 80.0 | 66.0 | 64.0 | 58.0 | 62.0 | 66.0 |
| 4 | ✓ | 76.0 | 70.0 | 56.0 | 64.0 | 62.0 | 65.6 |
| 5 | ✓ | 76.0 | 62.0 | 60.0 | 54.0 | 66.0 | 63.6 |

outperforming Vanilla RAG. Notably, $G^2$-Reader at $k = 5$ (86.3%) already surpasses Vanilla RAG at $k = 20$ (86.2%), suggesting that evolved structural representations in the Content Graph enable more precise evidence capture with minimal overhead.

Figure 2b compares accuracy across various model sizes, showing that $G^2$-Reader maintains a substantial lead over Single VLM baselines. Even with smaller open-source backbones, our system exceeds the performance of much larger standalone frontiers, demonstrating that the dual-graph architecture effectively compensates for the long-context reasoning limitations of medium-sized models.

### 4.5. Case Study

**Content Graph.** Figure 3 visualizes a subgraph from the Content Graph of a SciGraphQA sample. Each node corresponds to an atomic document element, while different colors indicate cliques formed by densely linked nodes. Notably, textual descriptions, mathematical formulations, and illustrative figures are tightly coupled within semantic neighborhoods, enabling effective graph-based evidence organization. This transparent representation also contributes to interpretability by revealing how retrieved evidence is grounded and relates to the original document structure. Please find more examples in Appendix D.1.

**Planning Graph.** Here, we present an example in Figure 4, illustrating how $G^2$-Reader constructs the planning graph and refine it. Initially, the Decomposer breaks down the root question into two parallel sub-questions. Subsequently, upon reviewing the Worker's output, the Evidence Checker identifies a discrepancy, flagging the gap: "Missing the data for Psalm 151 in the retrieved Table 1 and Table 2." Based on this feedback, the Decomposer initiates a replanning process for the planning graph; it retains node $n_1$ but updates node $n_2$ as $n_2^*$ and appends a new sub-node $n_3$, aiming to

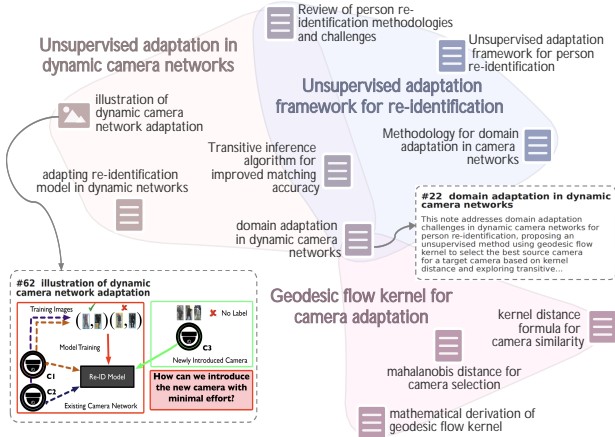

*Figure 3.* Multimodal Content Graph of a representative Sci-GraphQA example based on (Panda et al., 2017). Node summaries are further manually condensed and annotated for readability.

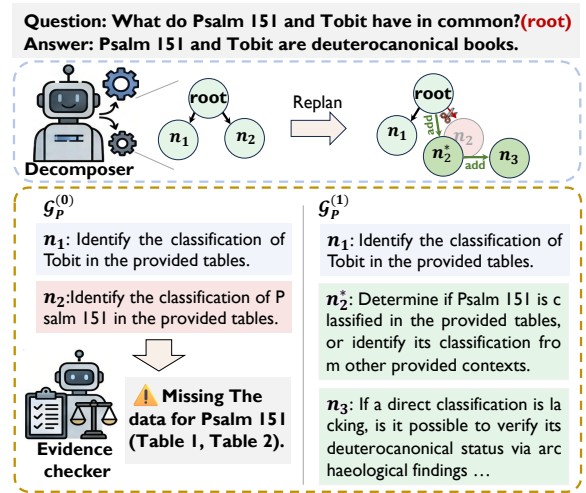

*Figure 4.* Example of Planning Graph Updating. Nodes updates and additions to bridge information gaps.

verify the result using materials beyond the provided tables. Ultimately, this adaptive workflow enables the system to bridge the evidence gap and correctly conclude that both are deuterocanonical books, thereby avoiding the potential failures inherent in static retrieval plans.

## 5. Conclusion

This paper studied multimodal long-document QA through the lens of two coupled limitations of current RAG systems: flat chunking fragments cross-modal evidence and breaks document-native structure, and long-context retrieval remains unstable even with iterative search, often looping or drifting without a reliable notion of progress. To address these issues, we proposed $G^2$-Reader, a dual-graph system that combines a Content Graph for grounded, structure-

preserving evidence representation with a Planning Graph that organizes reasoning as an agentic sub-question DAG to guide stepwise navigation and evidence assembly. Experiments on VisDoMBench across five multimodal domains show that $G^2$-Reader with an open-source backbone achieves the best overall performance (66.21%), outperforming strong baselines and a standalone GPT-5. Comprehensive ablations further confirm the effectiveness and complementarity of the Content Graph and Planning Graph components. Future work will explore more reliable relation induction, improved evidence sufficiency checks for planning, and more efficient inference-time navigation for broader real-world deployments.

## Impact Statement

This paper presents work whose goal is to advance the field of Machine Learning by improving retrieval-augmented reasoning over multimodal long documents. Our dual-graph formulation promotes a "documents-as-knowledge" perspective by turning visually rich documents into an explicit, reusable external knowledge structure that preserves cross-modal relations and supports evidence-based reasoning. This can facilitate practical user-side deployment of long-document capabilities and enable more controllable knowledge management (*e.g.*, selective updates and access control) compared to relying solely on parametric memory.

## Acknowledgments

This work was partly supported by the National Natural Science Foundation of China under grant No.92371206 (W.Y.) and the Intelligent Game and Decision Laboratory.

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

# A. Evaluation protocol

We provide comprehensive details about our evaluation methodology. The accuracy metric is computed using an automated LLM-based evaluator that performs semantic matching between generated answers and ground-truth responses, ensuring robustness to variations in phrasing while maintaining strict correctness criteria for factual content.

---

**Prompt: RAG Accuracy Evaluator**

You are an expert evaluator tasked with assessing the accuracy of answers generated by a RAG (Retrieval-Augmented Generation) system.

**Task**: Evaluate whether the generated answer correctly responds to the given question based on the expected answer.

**Question**: {question}
**Expected Answer**: {gold_answers}
**Generated Answer**: {assistant_answer}

**Evaluation Criteria**:
- **Accuracy (0 or 1)**: Does the generated answer match the factual content of the expected answer?
    - 1: The generated answer is factually correct and aligns with the expected answer.
    - 0: The generated answer is factually incorrect or contradicts the expected answer.

**Instructions**:
- Focus on factual correctness, not writing style or format.
- Consider partial matches: if the generated answer contains the correct information but includes additional context, it should still be considered accurate.
- For numerical answers, check if the values match or are equivalent.
- For list answers, check if all key elements are present.

**"Not answerable" handling**:
- If BOTH the expected answer AND generated answer indicate "Not answerable" or inability to answer, consider it accurate (accuracy=1).
- If the expected answer provides specific content but the generated answer says "Not answerable", it is INCORRECT (accuracy=0).
- If the expected answer says "Not answerable" but the generated answer provides specific content, it is INCORRECT (accuracy=0).

**Output Format**:
Please respond with a JSON object containing only:

```
{
  "accuracy": 0 or 1,
  "reasoning": "Brief explanation of your evaluation"
}
```

- Output MUST be valid JSON (double quotes only).
- Do NOT use LaTeX, math delimiters, or any backslashes "in reasoning.
- If you need to mention $N_C$, write it as "N_C" or "NC" (no "\" characters).
- No code fences, no extra text.

---

---

**Prompt: Recall Evaluator**

You are a harsh and rigorous Retrieval Evaluation Judge.

**Task**

Given (A) ground-truth "Evidence" (may be a text snippet or structured JSON) and (B) a "Retrieved Context", judge whether the context explicitly supports the evidence.

**Evidence formats you may see**
- A plain text excerpt from a paper (e.g., a sentence/paragraph).
- A table mention / caption text (e.g., "FLOAT SELECTED: Table 3: ...").
- A comma-separated key:value list (common in table-style datasets).
- A JSON object with fields (e.g., keys like "figure_caption" and "first_mention").

**How to judge (STRICT, DOMAIN-AGNOSTIC)**
1) Build a checklist of atomic "evidence claims" FROM THE EVIDENCE ITSELF.
    - Extract a checklist from the Evidence. Each checklist item MUST be directly grounded in the Evidence text/JSON.
    - Keep claims atomic: one key fact per item. Keep numbers/dates/names as-is.
2) For each claim, check if the Retrieved Context explicitly supports it.
    - Support must be explicit. If unsure, mark as missing.
    - If the context provides a conflicting value for the same fact, mark as contradicted.
3) Score STRICTLY:
    - 1.0 Full Support: ALL claims supported; no contradictions; critical numbers/dates/names match.
    - 0.7 Near Full Support: only 1 minor claim missing; no critical numbers/dates/names missing; no contradictions.
    - 0.3 Partial/Weak Support: multiple claims missing but at least some are clearly supported and same topic.
    - 0.0 No Support/Contradictory: none supported, OR any contradiction exists for a claim.

**Non-negotiable strictness**
- Do NOT infer or guess missing facts.
- Apply Zero-Knowledge Principle: use ONLY the Retrieved Context to decide support.

**Output (JSON only)**

Return ONLY a JSON object with these fields:

```
{
  "score": float,
  "label": "Full Support" | "Near Full Support" |
           "Partial/Weak Support" | "No Support/Contradictory",
  "supported_count": int,
  "total_segments": int,
  "missing_segments": array of strings,
  "contradicted_segments": array of strings,
  "reasoning": string
}
```

**# Inputs**

Evidence (raw):

{evidence_raw}

Retrieved Context:

{retrieved_context}

## B. Implementation details of baselines

**Single-VLM.** The Single VLM pipeline serves as the primary baseline for document-based visual question answering. This approach utilizes a single large-scale vision-language model to process input information in a zero-shot manner through a three-stage workflow.First, in PDF Parsing and Modality Extraction, we extract multimodal content from source documents $\mathcal{D}$ for a given query $q$. Automated tools decouple the documents into textual sequences $T$ and visual elements $I$, such as tables and figures, to preserve both semantic and visual evidence. Second, for Context Truncation and Sampling, we

apply heuristic strategies to meet the context window constraints of models. Textual content is partitioned into chunks and sampled to fit the input budget, while a fixed number of representative images are selected to maintain multimodal coverage without exceeding memory limits. Finally, during Joint Multimodal Inference, the sampled text, selected images, and query are interleaved into a single prompt. The model generates a response using a Chain-of-Thought (CoT) prompting style, providing a reasoning trace before the final answer.

**Deepseek-OCR.** The DeepSeek-OCR pipeline serves as an alternative baseline to evaluate the impact of structured document parsing on multimodal reasoning. This approach operates in a zero-shot, single-pass inference mode through a three-stage workflow, emphasizing structural integrity in its input representation.First, in Structured Document Parsing, we process source PDFs using DeepSeek-OCR to generate Multimodal Markdown (MMD) content. This process explicitly reconstructs hierarchical headings, LaTeX-formatted equations, and Markdown tables to provide an organized semantic layout. Second, for Token-Aware Context Truncation, the MMD content is trimmed using a binary search-based truncation method to fit the model's context window while preserving the most relevant prefixes. Finally, during Multimodal Joint Inference, the structured MMD text is interleaved with high-resolution visual elements extracted from the primary document. The model then generates a response following a Chain-of-Thought (CoT) reasoning path, leveraging the structured evidence to answer complex queries. **Rule-based Memory Evolution (*Lite* variant).** We implement a lightweight *rule-based* variant of $G^2$-Reader, which employs a different evolution operator while keeping the same initialization, retrieval, and planning components. Concretely, each node $v_i$ is represented by an embedding $\mathbf{h}_i$, initialized identically to the full variant. At each evolution iteration, node representations are updated via similarity-based propagation rather than VLM rewriting. Specifically, for each node, we compute cosine similarities $S_{ij} = \cos(\mathbf{h}_i, \mathbf{h}_j)$ and select the top-$K$ most similar neighbors $\mathcal{N}_i$. The updated embedding is obtained by similarity-weighted averaging:

$$\mathbf{h}_i^{(t+1)} = \alpha \mathbf{h}_i^{(t)} + (1 - \alpha) \frac{\sum_{j \in \mathcal{N}_i} S_{ij} \mathbf{h}_j^{(t)}}{\sum_{j \in \mathcal{N}_i} S_{ij}}.$$

Graph edges are deterministically updated to connect $v_i$ with $\mathcal{N}_i$. Unlike the full variant, *Lite* does not regenerate summaries, keywords, or semantic relations, and all evolution is performed in embedding space without LLM calls. This design significantly reduces evolution cost while sacrificing part of the performance due to lack of semantic abstraction and relation inference enabled by VLM-based evolution.

## C. Efficiency Analysis

We analyze the computational cost of Content Graph construction, comparing the full VLM-based evolution with the lightweight rule-based variant (*Lite*). Table 5 reports the average per-document cost across 50 samples, covering token usage, monetary cost, and wall-clock time. Note that although the open-source Qwen3-VL-32B-Instruct model is deployed locally, we still estimate the monetary cost based on publicly available API pricing from OpenRouter, a third-party model routing service, to provide a practical cost reference.

*Table 5.* Efficiency comparison of Content Graph construction. The *Lite* variant replaces VLM-based memory evolution with rule-based updating, eliminating LLM calls during the evolution phase. Savings are computed relative to the original configuration.

| Configuration | Token Usage | Token Cost | Time (s) |
|---|---|---|---|
| Original (VLM-based) | 2,174,531 | $1.22 | 233.6 |
| Lite (Rule-based) | 471,118 | $0.32 | 130.4 |
| **Saving** | **−78.3%** | **−73.7%** | **−44.2%** |

As shown in Table 5, the *Lite* variant achieves substantial efficiency gains by replacing VLM-based memory evolution with a lightweight rule-based approach. Specifically, since the evolution phase no longer requires LLM calls, token consumption is reduced by 78.3% (from 2.17M to 471K tokens per document), and monetary cost decreases by 73.7% (from $1.22 to $0.32). Wall-clock time is also reduced by 44.2%, as embedding-based similarity computation is significantly faster than iterative VLM inference.

Despite these efficiency gains, the *Lite* variant maintains competitive performance, as demonstrated in Table 4: at $T{=}3^\star$, it achieves 67.0% average accuracy compared to 68.0% for full VLM-based evolution—a margin of only 1.0%. This favorable accuracy–efficiency trade-off makes the *Lite* variant particularly suitable for cost-sensitive or latency-critical deployments,

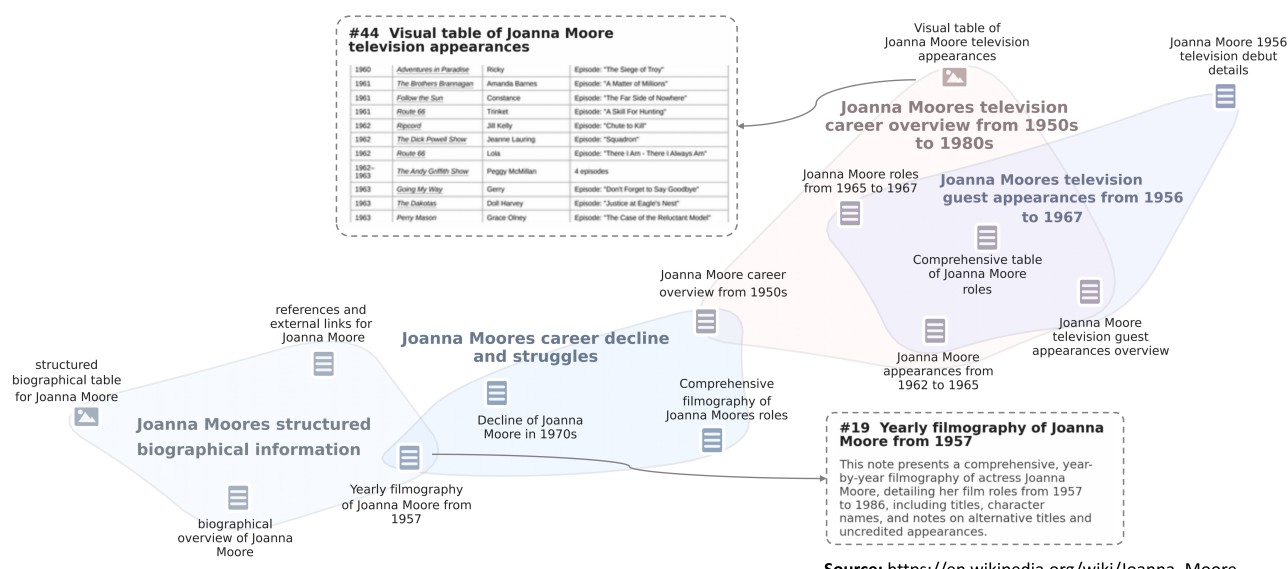

*Figure 5.* Content Graph of Wikipedia webpage from **FetaTab**

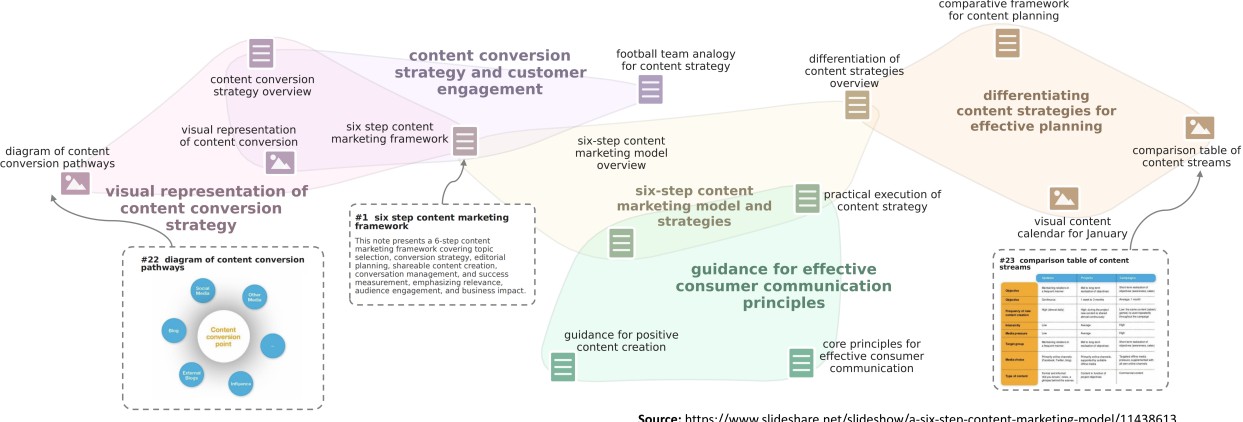

*Figure 6.* Content Graph of slide decks from **SlideVQA**

where the modest performance reduction is outweighed by the significant computational savings.

## D. Examples

### D.1. Content Graph

We present additional visualizations of the Content Graphs constructed by $G^2$-Reader on diverse document types. Figure 5 and Figure 6 showcase representative subgraphs from FetaTab and SlideVQA, respectively. Despite substantial differences in document structure and modality, $G^2$-Reader consistently organizes heterogeneous elements–such as text passages, tables, figures, and diagrams–into coherent graph neighborhoods, demonstrating its superior generality.

### D.2. Planning Graph

This section provides a concrete demonstration of how $G^2$-Reader dynamically optimizes its planning graph ($\mathcal{G}_P$) through three iterations, using a complex case study from PaperTab in VisDoMBench. The task requires the system to identify a specific paper (concerning a news-driven recurrent regime-switching model) and calculate the total number of documents within its dataset. To ensure conciseness, for the second and third iterations, we present only the retrieval results and logical

shifts associated with the updated nodes.

## E. Prompts

In this section, we present prompts used in $G^2$-Reader, including Initial Decomposer which is used to construct initial planning graph (E.1), Refinement Decomposer which is used to refine the planning graph (E.2), Worker (E.3), Reasoner (E.4), Evidence Checker (E.5), Content Graph node initialization (E.6) and evolution (E.7).

### E.1. Prompts Template of Initial Decomposer

You are an expert task decomposition assistant. Your goal is to break down the given question into a Directed Acyclic Graph (DAG) for structured, step-by-step reasoning.
**Key Rules:**
- **Structure:** Must be a Directed Acyclic Graph (DAG). The root node must be the original question.
- **Dependency:** The resolution of a child node's task must depend on its parent.
- **Atomicity:** Each node must represent a minimal, indivisible sub-task. Focus only on extracting the essential facts, relationships, or calculations from the provided context (including any textual or visual elements) required to solve the problem. Avoid irrelevant details or speculation.
- **Brevity:**
    - Limit the total number of nodes to 6 or fewer.
    - Limit the maximum depth of the graph to 3 levels (the root is level 1).

**Context:** $DOC$
**Original Question:** $Q$
Your response MUST strictly follow this format:
<dag>
{
    "nodes": [
        {
            "id": "root",
            "task": "The full original question.",
            "type": "question",
            "children": ["n1", "n2"]
        },
        {
            "id": "n1",
            "task": "A specific, atomic sub-task.",
            "type": "sub-question",
            "children": [ ]
        }
    ],
    "edges": [
        {"from": "root", "to": "n1"},
        {"from": "root", "to": "n2"}
    ]
}
</dag>
Now, decompose the given question into a DAG.

**E.2. Prompts Template of Refinement Decomposer**

You are an expert task decomposition assistant. Your goal is to refine the given question into a Directed Acyclic Graph (DAG) for structured, step-by-step reasoning. This DAG enables layered, progressive inference.

This is an adjustment round: Use the original question, context, current evidence, previous DAG, and identified gaps to perform targeted CRUD operations (Create: add nodes for gaps; Read: reference existing nodes/evidence; Update: refine tasks/edges; Delete: remove redundants/irrelevants). Generate a minimal DAG that changes the previous one by net +1-3 nodes, preserving useful structure while resolving gaps—prioritize atomic tasks, avoid over-decomposition or unrelated details.

**Key Rules:**
- **DAG structure:** Must be acyclic and directed (dependencies from parent to children).
- **Root node:** Unchanged original question.
- **Nodes:** Atomic sub-tasks (self-contained, evidence-aligned, derived from the provided context, which may include both textual and visual elements); total $\leq 8$, depth $\leq 3$ (merge/prune as needed).
- **Interdependencies:** Children resolve only after parents.
- **CRUD focus:** Scan previous DAG/evidence for reusability; add/update/delete minimally per gap.

**Context:** $DOC$
**Original Question:** $Q$
**Previous DAG:** $OLD\_DAG$ (JSON; base refinements here)
**Current Evidence (QA pairs):** $EVIDENCE$
**Identified Gaps:** $GAPS$ (If none, lightly refine, e.g., merge redundants.)
**Node format:**
- "id": Unique string (reuse from previous DAG where possible).
- "task": Concise, atomic description (refine via evidence/gaps).
- "type": "question" for root, "sub-question" for others.
- "children": Array of child IDs (empty [ ] for leaves).

**Your response MUST follow this format:**
```
<dag>
{
    "nodes": [
        {
            "id": "root",
            "task": "Full original question here.",
            "type": "question",
            "children": ["n1", "n2"]
        },
        {
            "id": "n1",
            "task": "Atomic sub-task.",
            "type": "sub-question",
            "children": ["n3"]
        }
        ...
    ],
    "edges": [
        {"from": "root", "to": "n1"},
        {"from": "root", "to": "n2"}
        ...
    ]
}
</dag>
```
Now, adjust the DAG for the given question.

**E.3. Prompts Template of Worker**

Please read the following retrieved text chunks and any accompanying images (if provided), then answer the question below.
<text>
$DOC$
</text>
<images>
If one or more images are provided, analyze their content and cross-reference them with the text; otherwise, rely entirely on the textual context
</images>
What is the only correct answer to this question: $Q$
**Guidelines:**
- Base your reasoning strictly on the provided text and images (if any): Cite specific chunks (e.g., "From Related Memory [1]: ...") or image elements (e.g., "From Image 1: The chart shows...") to ground claims; avoid hallucinations or external knowledge.
- Think step-by-step: Identify key facts from text and/or images, relate to question, derive logical conclusion.
- If text/images are insufficient, note limitations but provide best grounded inference.
- Integrate multimodal evidence: Cross-reference visuals with text for deeper insights (e.g., "The image diagram confirms the text description in Related Memory [2]").
<thought>
[your step-by-step thought process and grounded conclusion]
</thought>
<output>[Your Final Answer]</output>

**E.4. Prompt Template of Reasoner**

You are a helpful AI assistant that excels at question answering. Your task is to answer the following question based on the given context and any accompanying images (if provided). To help you better generate the correct answer, some relevant Q&A documents that provide supplementary information from sub-tasks will be provided.

Also, to help you better answer the question, please follow these steps:

1. First, carefully read and understand the question.
2. Then, thoroughly analyze the provided context and images (if any): Describe key visual elements from images and cross-reference them with text for comprehensive insights. If no images, focus on text alone.
3. Integrate insights from the supplementary Q&A trajectory, noting any visual-text alignments.
4. Think step by step to arrive at your answer, ensuring it is concise, factual, and directly supported by the evidence (text + images + trajectory).
5. Generate exactly ONE complete answer as a short text response.

**QUESTION (Please read this carefully):**

$Q$

**RELEVANT CONTEXT (raw):**

$DOC$

**RELEVANT IMAGES (if provided):**

If one or more images are provided, analyze their content and cross-reference them with the text; otherwise, rely entirely on the textual context

**RELEVANT Q&A TRAJECTORY (sub-task insights):**

$TRA$

**Your response MUST follow this format:**

<thought>

1. [First step of your reasoning, referencing key context and/or images]

2. [Second step of your reasoning, integrating trajectory insights and multimodal evidence]

...

[Conclusion: Summarize the evidence leading to the final answer]

</thought>

<output>Your complete answer here (concise text)</output>

**E.5. Prompt Template of Evidence Checker**

You are an expert evidence checker for structured reasoning tasks. Your role is to evaluate whether the provided evidence from sub-tasks and initial retrieval is sufficient to comprehensively answer the original question.

**Key Guidelines:**

- **Sufficiency means:** The combined evidence covers all key facts, relationships, calculations, and insights needed for a complete, accurate answer without major gaps (e.g., gaps $<= 1$ minor issue).
- **Consider coverage:** Does it address the core aspects of the question? Are there unresolved sub-problems, missing data, or ambiguities? Be conservative: If gaps $> 1$ or any critical aspect is unclear/incomplete, mark as insufficient.
- **Prioritize atomic gaps:** List specific, actionable sub-problems (e.g., "Missing calculation for Z based on fact Y").

**Inputs:**

Original Question: $Q$

Initial Retrieved Context: $DOC$

Sub-task Evidence (QA pairs from DAG nodes): $EVIDENCE$

Output ONLY a valid JSON object wrapped in <check> tags. No explanations.

**Format:**

<check>
{
    "sufficient": true, // or false
    "gaps": [ ] // If false, list 1-3 specific gaps as strings (e.g., ["Missing sub-problem: Compute impact of X on Y"]);
if true, empty array
}
</check>

If insufficient, the system will use this to trigger DAG adjustment.

**E.6. Prompt Template of Content Graph Node Initialization**

Generate a structured analysis of the following content by:
1. Identifying the most salient keywords (focus on nouns, verbs, and key concepts)
2. Extracting core themes, concepts and arguments
3. Creating relevant categorical tags

Note that when the provided content contains no meaningful information, such as reference lists, the summary you generate must include exactly `"No meaningful information"` to aid filtering.

Format the response as a JSON object with the following structure:
{
 "keywords": [
  // several specific, distinct keywords that capture key concepts and terminology
  // Order from most to least important
  // At least three keywords, but don't be too redundant.
 ],
 "summary":
  // one sentence summarizing:
  // - Main topic/domain
  // - Key arguments/points
  // - Be concise, informative and to the point.
 ,
 "tags": [
  // several broad categories/themes for classification
  // Include domain, format, and type tags
  // At least three tags, but don't be too redundant.
 ]
}

**Content for analysis:**
{document chunk}

**E.7. Prompt Template of Content Graph Evolution**

You are an AI memory evolution agent responsible for managing and evolving a knowledge base. Here is a memory note (including its content, summary, and keywords), and a few neighboring notes. Make decisions about its evolution.
**The memory note content:** {content}
**Summary:** {context}
**Keywords:** {keywords}
**The {neighbor_number} neighboring notes:**
{neighbors}
Based on this information, determine:
1. Which of the neighboring notes should be linked to this memory note?
2. Should the summary and keywords of this memory note be updated, considering its relationships with the neighboring notes?
3. If so, what should be the new summary and keywords?

If you determine that the memory note needs not be updated, the new summary and keywords should be the same as the original summary and keywords. Please keep the summary concise, informative and to the point, with no more than 30 words.
**Definition of Valid Edge Relationships:**
Two memory notes should be connected ONLY if they exhibit one or more of the following:
  - **Direct Reference/Citation**; **Causal Relationship**; **Part-Whole Relationship**; **Conceptual Elaboration**; **Temporal Sequence**; **Contrastive/Comparative**; **Hierarchical Relationship**; **Contextual Dependency**.
**Summary and Keywords Update Principles:**
  - Use neighbors as reference, not as content; Focus on unique aspects; Distinctive keywords; Self-contained description.
Return your decision in JSON format with the following structure:
{
    "suggested_connections": ["neighbor_memory_ids"],
    "should_update": True or False,
    "new_summary": "new summary",
    "new_keywords": ["keyword_1", ..., "keyword_n"]
}

## Planning Graph

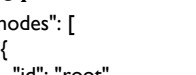

$\mathcal{G}_P^0$

**Decomposer**

```
"nodes": [
  {
    "id": "root",
    "question": "What is the total number of documents in the datas
et in the paper on modeling stock movements using a recurrent state
transition model influenced by news events?",
    "children": ["n1","n2"]
  },
  {
    "id": "n_1",
    "subquestion": "Identify the paper that models stock movements
using a recurrent state transition model influenced by news events.",
    "children": [ ]
  },
  {
    "id": "n_2",
    "subquestion": "Determine the total number of documents in the
dataset used in the identified paper.",
    "children": [ ]
  }]
```

## Evidence Check

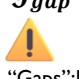

$\mathcal{G}_{gap}^0$

**Evidence checker**

⚠️

"Gaps":[

"The provided context does not contain any information abo
ut the dataset size for the ANRES model described in the pap
er on modeling stock movements using a recurrent state tran
sition model influenced by news events. The dataset size men
tioned (26,892 documents) refers to CAIL2019-SCM, which i
s unrelated to the ANRES model.",

"There is no mention of the number of trading days, samples,
or documents in the dataset used for training or evaluating th
e ANRES model in the retrieved text or images."]

## Retrieval Results

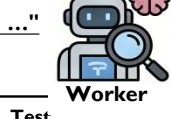

**Worker**

**$n_1$ : "Identify the paper that models stock ..."**

**Related content[0]:**
…ANRES uses LSTM…The underlying stock value trends are represent
ed as a recurrent state $z$ transited over time…In each trading day, w
e consider the impact of corresponding news events…

$$z_t' = \overline{LSTM}(v_t, z_{t-1})$$
$$z_t = f(z_t')$$

where $v_t$ is the news events impact vector…Then we use the sequentia
l state $z_t$ to make binary classification…

**Related content[1]:**
…we use scaled dot-product attention to capture the influence of new
s over a period for the recurrent state transition…we first transform t
he last trading day's stock value $z_{t-1}$ to a query vector $q_t$ :

$$q_t = \tanh(W^q z_{t-1})$$
$$\gamma_t = softmax\left(\frac{C_t^i q_t}{\sqrt{V}}\right)$$
$$\beta_t = softmax\left(\frac{B_t^i q_t}{\sqrt{V}}\right)$$

We sum the news events embedding matrices to obtain news events i
mpact vectors $c_t$ and $b_t$…

**Related content[2]:**
…obtain the input $v_t$ for LSTM-based recurrent state transition:

$$v_t = \begin{cases} [c_t, b_t, a_t], & \text{when training} \\ [c_t, b_t, \hat{a}_t], & \text{when evaluatin} \end{cases}$$

In this model, all other factors…are explicitly modeled as noise using a
random factor…

**Related content[3]:**
…obtain the noisy recurrent state $z_t$ as:

$$\epsilon_t \sim \mathcal{N}(0,1)$$
$$z_t = tanh(z_t' + \sigma_t \epsilon_t)$$

We use the public financial news dataset…predicting the Standard &
Poor's 500 stock (S&P 500) index and its selected individual stocks…

**Related content[4]:**

### News-Driven Stock Prediction With Attention-Based Noisy Recurrent State Transition

Xiao Liu, Heyan Huang, Yue Zhang, Changsen Yuan
School of Computer Science and Technology, Beijing Institute…

**$n_2$ : "Determine the total number of docu ..."**

**Related content[0]:**

**Table 1: Statistics of the datasets**

| | Training | Development | Test |
|---|---|---|---|
| #documents | 358,122 | 96,299 | 99,030 |

**Related content[1]:**
More specifically, CAIL2019-SCM contains 8,964 triplets of legal d
ocuments…By detecting similar cases in triplets, we can apply this
algorithm for ranking all documents to find the most similar docu
ment in the database…

**Related content[2]:**
CAIL2019-SCM: A Dataset of Similar Case Matching in Legal Domain
Abstract
… CAIL2019-SCM contains 8,964 triplets of cases published by the Su
preme People's Court of China…

**Related content[3]:**
…The input of CAIL2019-SCM is a triplet $(A, B, C)$, where
$A, B, C$ are fact descriptions of three cases… the task of CAIL201
9-SCM is to predict whether $sim(A, B) > sim(A, C)$ or
$sim(A, C) > sim(A, B)$.

**Related content[4]:**
…First, we select many documents within the range of Private Len
ding…To produce qualified triplets, we first annotated some cruci
al elements in Private Lending for each document…After annotati
ng these elements, we can assume that cases with similar elements
are quite similar…We have constructed 8,964 triples in total by t
hese methods, and the statistics can be found from Table 1…Furt
hermore, to ensure the quality of annotation, every document and
triplet is annotated by at least three legal professionals to reach an
agreement.

*Figure 7.* Iteration 1 of Planning Graph

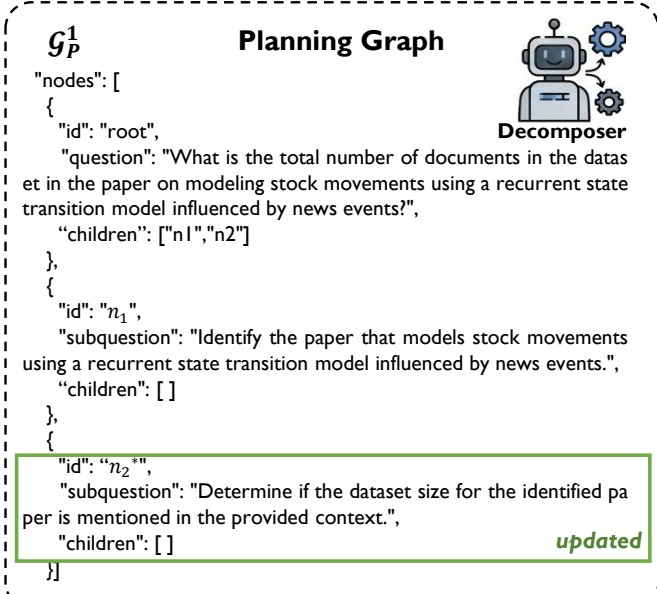

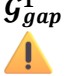

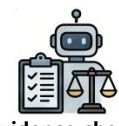

**Planning Graph** — Decomposer

$\mathcal{G}^1_P$

"nodes": [
  {
    "id": "root",
      "question": "What is the total number of documents in the dataset in the paper on modeling stock movements using a recurrent state transition model influenced by news events?",
      "children": ["n1","n2"]
  },
  {
    "id": "$n_1$",
      "subquestion": "Identify the paper that models stock movements using a recurrent state transition model influenced by news events.",
      "children": [ ]
  },
  {
    "id": "$n_2$*",
      "subquestion": "Determine if the dataset size for the identified paper is mentioned in the provided context.",
      "children": [ ]          *updated*
}]

**Evidence Check** — Evidence checker

$\mathcal{G}^1_{gap}$

⚠️

"Gaps":[
"Missing dataset size for the ANRES model: The provided context discusses datasets like IAM-OnDB, IBM-UB-1, and 'own dataset', but does not explicitly state the size of the dataset used for training or evaluating the ANRES model.",
"Ambiguity in dataset reference: The context mentions multiple datasets (e.g., IAM-OnDB, IBM-UB-1, own dataset) without clearly linking any to the ANRES model described in Related Memory [0], making it impossible to determine which dataset's size is relevant to the question."]

**Retrieval Results(updated nodes)**
$n_2$ : "Determine if the dataset size for the…"

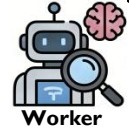
Worker

**Related content[0]:**

Table 1: Statistics of the datasets

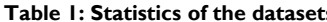

| | Training | Development | Test |
|---|---|---|---|
| #documents | 358,122 | 96,299 | 99,030 |

**Related content[1]:**
… In addition, we show the relative improvement in error rates on the languages for which we have evaluation datasets of more than 2 000 items (Figure 7). The new architecture performs between 2 0% - 4 0% (relative) better over almost all languages …

**Related content[2]:**
We want to highlight the fundamental differences between the different datasets.
Table 8 CER comparison when training and evaluating IAM-OnDB, IBM-UB-1 and our Latin training/eval set.

| train \ test | IAM-OnDB | IBM-UB-1 | own dataset |
|---|---|---|---|
| IAM-OnDB | 3.8 | 17.7 | 31.2 |
| IBM-UB-1 | 35.1 | 4.1 | 32.9 |
| own dataset | 3.3 | 4.8 | 8.7 |

**Related content[3]:**
…The number of training samples varies from tens of thousands to several million per script, depending on the complexity and usage. We provide more information about the size of our internal training and tests datasets in table 9…However, overfitting is less pronounced probably because our datasets are substantially larger than the publicly available datasets…

**Related content[4]:**
… Another publicly-accessible English-language dataset is the IBM-UB-1 dataset [43]. From the available datasets therein, we use the English query dataset, which consists of 63 268 handwritten English words…We split this dataset into 4 parts with non-overlapping writer IDs: 47 108 items for training, 4 690 for decoder weight tuning, 6 134 for validation and 5 336 for testing7…

*Figure 8.* Iteration 2 of Planning Graph

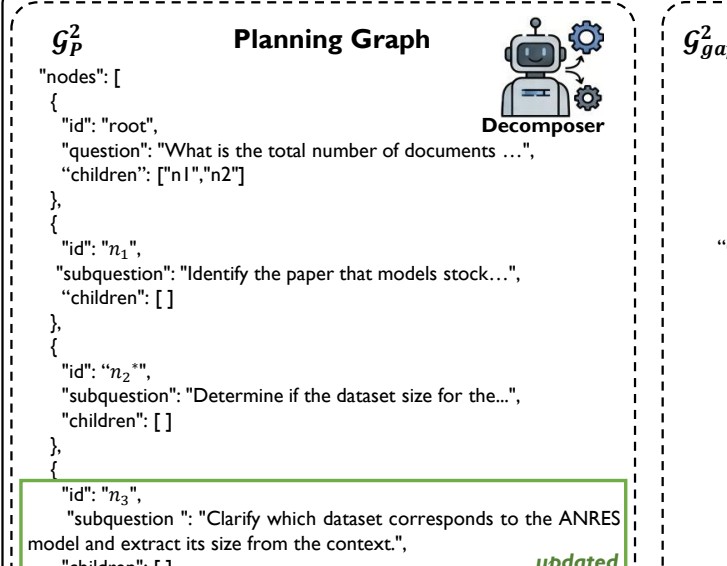

**Planning Graph**

$\mathcal{G}_P^2$

**Decomposer**

```
"nodes": [
  {
    "id": "root",
    "question": "What is the total number of documents …",
    "children": ["n1","n2"]
  },
  {
    "id": "n_1",
    "subquestion": "Identify the paper that models stock…",
    "children": [ ]
  },
  {
    "id": "n_2*",
    "subquestion": "Determine if the dataset size for the…",
    "children": [ ]
  },
  {
    "id": "n_3",
    "subquestion": "Clarify which dataset corresponds to the ANRES
model and extract its size from the context.",
    "children": [ ]                                              updated
  }
]
```

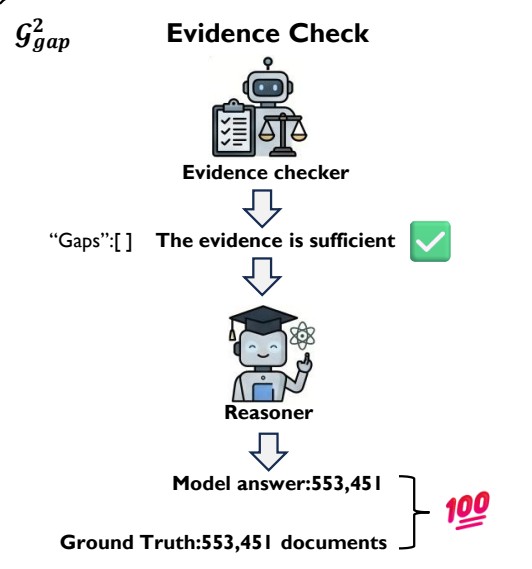

**Evidence Check**

$\mathcal{G}_{gap}^2$

**Evidence checker**

"Gaps":[ ]    The evidence is sufficient ✅

**Reasoner**

Model answer:553,451

Ground Truth:553,451 documents

**Retrieval Results(updated nodes)**

$n_3$: "Clarify which dataset corresponds to the ANRES …"

**Worker**

**Related content[0]:**

… Following previous work [20, 3, 21], we adopt the standard measure of accuracy and Matthews Correlation Coefficient (MCC) to evaluate S&P 500 index prediction and selected individual stock prediction …

**Table 2: Hyper-parameters setting.**

| Name | Value |
|------|-------|
| …… | …… |
| news embedding dimension V | 256 |
| recurrent state dimension D | 256 |
| trading sequence length T | 100 |

… For each trading day, we compare the results whether states transitions are modeled or not … In summary, the following four baselines are designed: … ANRES_Sing_R … ANRES_Sing_Z … ANRES_Seq_R … ANRES_Seq_Z …

**Related content[1]:**

Table 1: Statistics of the datasets.

| | Training | Development | Test |
|---|---|---|---|
| #documents | 358,122 | 96,299 | 99,030 |
| #samples | 1,425 | 169 | 191 |
| Time span | 10/20/2006-06/18/2012 | 06/19/2012-02-21/2013 | 02/22/2013-11/21/2013 |

…We use the public financial news dataset released by [3], which is crawled from Reuters and Bloomberg over the period from October 2006 to November 2013. We conduct our experiments on predicting the Standard & Poor's 500 stock (S&P 500) index and its selected individual stocks…

**Related content[2]:**

…Development set results on predicting S&P 500 index are shown in Table 3…We use the development set to find a suitable length T for trading sequence, which is searched from { 1 , 3 , 5 , 7 , 9 , 11 , 13 , 15 } …we choose the hyper-parameter T = 7 and use it in the remaining experiments.

**Related content[3]:**

…In this paper, we define future news events as those that are published within seven calendar days after the trading day $t$ …We concatenate the above-mentioned three types of news events impact vectors to obtain the input $v_t$ for LSTM-based recurrent state transition on trading day $t$ as:

$$v_t = \begin{cases} [c_t, b_t, a_t], & \text{when training} \\ [c_t, b_t, \hat{a}_t], & \text{when evaluatin} \end{cases}$$

**Related content[4]:**

**Table 2: Hyper-parameters setting.**

| Name | Value | | Name | Value |
|------|-------|---|------|-------|
| batch size | 16 | | MSE loss weight θ | 0.4 |
| learning rate lr | 0.005 | | regularization weight λ | 0.0005 |
| SGD momentum μ | 0.9 | | news embedding dimension V | 256 |
| dropout rate r | 0.3 | | recurrent state dimension D | 100 |
| | | | trading sequence length T | 7 |

*Figure 9.* Iteration 3 of Planning Graph

