# OpenReview forum: "$G^2$-Reader: Dual Evolving Graphs for Multimodal Document QA"
_ICML.cc/2026/Conference — ICML 2026 regular_

### Official Review · Reviewer_qh1o · 2026-03-07

**Soundness:** 3
**Presentation:** 3
**Significance:** 3
**Originality:** 3
**Overall Recommendation:** 4
**Confidence:** 4

**Summary:**

This paper introduces G2-Reader, a dual-graph framework designed to improve retrieval-augmented generation (RAG) for multimodal long-context question answering. A static, heterogeneous graph built from documents by parsing multimodal elements  into atomic nodes. Each node contains raw content, a summary, keywords, and embeddings. The graph undergoes iterative joint evolution where nodes refine their summaries and connections based on semantic neighbors, preserving structural and cross-modal dependencies. A dynamic directed acyclic graph (DAG) that explicitly models the reasoning process for a given query. It is constructed by a "decomposer" that breaks the query into atomic sub-questions with dependency edges. A "worker" retrieves evidence subgraphs from the Content Graph for each node, and an "evidence checker" evaluates sufficiency. If gaps exist, the Planning Graph is iteratively refined until evidence is complete. A final "reasoner" synthesizes the answer.

**Compliance With Llm Reviewing Policy:**

Affirmed.

**Key Questions For Authors:**

See the weaknesses.

**Limitations:**

See the weaknesses.

**Strengths And Weaknesses:**

Strengths:

S1: The decomposition into a static Content Graph and a dynamic Planning Graph  is elegant and principled. This separation allows the system to preserve document structure while enabling flexible, query-specific reasoning. The Planning Graph as a DAG ensures well-defined execution order and prevents cycles.

S2: The Content Graph evolves through joint updates of node attributes and topology, using VLMs to enrich summaries and infer meaningful relations  . This moves beyond simple similarity-based graphs. Ablations  show that evolution up to 3 rounds significantly improves accuracy  , and the structured subgraph readout  further boosts performance, especially for fragmented evidence.

Weaknesses：

W1: This complexity may limit adoption in resource-constrained settings and raises reproducibility concerns if specific VLM versions (e.g., GPT-4o, Qwen3-VL) are not publicly available or change over time.

W2: While the paper shows overall gains, it does not systematically analyze where G2-Reader still fails. For example, the Planning Graph refinement process could introduce errors if the decomposer generates misleading sub-questions or the evidence checker misjudges sufficiency. A breakdown of error types (e.g., retrieval failure, decomposition error, reasoning error) would strengthen the paper and guide future improvements.

---

> ### Author Rebuttal · Authors · 2026-03-31
>
> We sincerely appreciate your time and thoughtful comments. Below, we respond to each point in detail.
>
> ---
>
> **W1-1: This complexity may limit adoption in resource-constrained settings.**
>
> 1. **$G^2$-Reader admits a lower-cost operating point for resource-constrained deployment.**
> In addition to the full VLM-based pipeline, we evaluate a cheaper rule-based Content Graph construction variant. It reduces the one-time preprocessing overhead by roughly 3.8× while only causing a 1.0-point drop in final accuracy (68.0% → 67.0%), indicating a practical cost–performance tradeoff.
> 2. **The main extra cost is a one-time corpus-level build cost, not a recurring per-query burden.**
> Once built, the Content Graph is persistent and reusable across subsequent queries / sessions on the same corpus; therefore, the recurring deployment cost is only the online Planning/Inference stage.
> 3. **The recurring per-query cost remains moderate relative to the accuracy gain.**
> As shown below, the recurring per-query cost of G²-Reader is slightly cheaper than GPT-5 (0.0293 vs. 0.0318 dollar/query) while being substantially more accurate (68.00% vs. 51.92%). It is also far more cost-efficient than MA-RAG ($0.19/query). For one-off QA on a fresh corpus, we agree that the first-use cost is higher, and we will state this trade-off explicitly in the revision.
>
> | Method | Acc. (%) | Tokens | Time (s) | API cost($) |
> |:---:|:---:|:---:|:---:|:---:|
> | GPT-5 (single VLM) | 51.92 | 15,291.25 | 131.66 | 0.0318 |
> | Qwen3-VL-32B (single VLM) | 28.39 | 32,261.43 | 102.08 | 0.0163 |
> | MA-RAG | 34.28 | 386,134.13 | 103.37 | 0.19 |
> | **Ours** | 68.00 | 54,368.66 | 174.18 | 0.0293 |
>
>
>
> **W1-2: reproducibility concerns if specific VLM versions (e.g., GPT-4o, Qwen3-VL) are not publicly available or change over time.**
>
> 1. **In our main setup, the backbone is the open-source Qwen3-VL-30B-Instruct.** Thus, the method is not coupled to one proprietary API feature or one specific hosted model version.
> 2. **The evolved graph is reasonably stable across repeated stochastic builds.** We rebuilt the full Content Graph 3 times per query under T = 0.7 on a stratified sample from all five VisDoMBench subsets (current results: 38/50 completed queries). The graph remains stable across runs:
>
> | Metric | Mean |
> |:---:|:---:|
> | Keyword Jaccard$\uparrow$ | 0.453 |
> | Tag Jaccard$\uparrow$ | 0.726 |
> | Degree-distribution JS divergence$\downarrow$ | 0.032 |
>
> **W2: While the paper shows overall gains, it does not systematically analyze where G2-Reader still fails**
>
> We agree that a stage-wise failure analysis would strengthen the paper.
> To address this, we add an architecture-aligned error taxonomy for incorrect predictions, including:
> - **retrieval failure**, where the correct evidence is not retrieved or not sufficiently assembled;
> - **decomposition failure**, where the decomposer generates misleading or incomplete sub-questions;
> - **replanning failure**, where the evidence checker makes an incorrect sufficiency judgment and the refinement loop terminates too early or expands in the wrong direction; and
> - **final reasoning failure**, where the correct evidence is available but the final answer is still composed incorrectly.
>
> | Error type | Proportion among failures |
> |:---:|---|
> | **Retrieval failure** | 42.90% |
> | **Final reasoning failure** | 34.70% |
> | **Decomposition failure** | 18.40% |
> | **Replanning failure** | 4.10% |
>
> This analysis shows that (1) the dominant remaining bottlenecks are retrieval/evidence coverage and final reasoning, (2) while failures introduced specifically by replanning / sufficiency checking are present but not the primary source of error. We will include this breakdown and representative failure cases in the revision to better characterize the current limitations of G2Reader and guide future improvements.
>
> ---
> **Finally**: We really appreciate your insightful suggestions, which have already helped us strengthen both the analysis and presentation of this work. We hope the above clarifications resolve the concerns. If the response is helpful, we would be grateful if you could consider adjusting your score accordingly.

---

> > ### Author Rebuttal · Reviewer_qh1o · 2026-04-05
> >
> > Thanks for your responses. No more questions.

---

> > > ### Author Response · Authors · 2026-04-07
> > >
> > > Thank you for the thoughtful acknowledgement and for taking the time to review our rebuttal and additional experiments.

---

### Official Review · Reviewer_oqR4 · 2026-03-12

**Soundness:** 3
**Presentation:** 3
**Significance:** 2
**Originality:** 2
**Overall Recommendation:** 4
**Confidence:** 3

**Summary:**

The paper proposes G^2-Reader, a dual-evolving retrieval and reasoning framework for retrieval-augmented generation (RAG). The key idea is to iteratively refine both the retrieved context and the generated reasoning process. Specifically, the method alternates between retrieval and generation stages: the model first retrieves documents, generates intermediate reasoning, and then updates the retrieval query based on the evolving reasoning state. This dual-evolving process aims to progressively improve both the relevance of retrieved evidence and the quality of reasoning. The authors evaluate the method on several knowledge-intensive QA benchmarks and compare it with existing RAG-based approaches. Experimental results suggest that the proposed framework improves answer accuracy compared with several baselines.

**Compliance With Llm Reviewing Policy:**

Affirmed.

**Final Justification:**

I agree that most of my concerns are resolved during rebuttal. However, I still find the novelty is limited. the approach is more like a principled integration and refinement of existing ideas rather than a fundamentally new paradigm.

**Key Questions For Authors:**

1. How does the proposed dual-evolving framework differ fundamentally from iterative retrieval or multi-hop RAG approaches?
2. Can the authors provide analysis of how retrieval quality changes across iterations?
3. How robust is the approach to errors in intermediate reasoning that may affect query updates?
4. Which component contributes most to the observed improvements: iterative retrieval or other architectural choices?

**Limitations:**

Yes

**Strengths And Weaknesses:**

Strengths

The paper addresses an important challenge in retrieval-augmented generation: the mismatch between static retrieval results and evolving reasoning during multi-step inference. The proposed framework introduces an intuitive mechanism that allows retrieval and reasoning to interact iteratively, which may improve the relevance of retrieved information during inference. The method is conceptually simple and can be integrated with existing RAG architectures without major architectural changes. The empirical evaluation includes several QA benchmarks and compares against standard RAG baselines, suggesting improvements in answer accuracy. The paper also provides ablation studies examining the impact of iterative retrieval updates.

Weaknesses
1. The core idea of iterative retrieval during reasoning has appeared in prior work on multi-hop retrieval and iterative RAG frameworks. The paper does not clearly articulate what fundamentally distinguishes the proposed approach from existing iterative retrieval methods.

2. The method is primarily a procedural framework rather than a new model or learning formulation. The paper does not provide theoretical analysis or formal justification for why the dual-evolving process should improve retrieval or reasoning quality.

3. The retrieval update mechanism appears heuristic and depends on intermediate generated text, which may propagate reasoning errors. The paper provides limited analysis of error propagation or robustness of the iterative process.

4. The evaluation focuses on answer accuracy but provides limited analysis of retrieval behavior, such as how retrieval quality evolves across iterations or how often retrieval updates actually improve evidence relevance.

5. The empirical improvements over baselines appear moderate, and it is unclear whether the gains primarily come from iterative retrieval itself or from other implementation details.

---

> ### Author Rebuttal · Authors · 2026-03-31
>
> We sincerely appreciate your time and thoughtful comments. Below, we respond to each point in detail.
>
> **W1/Q1/Q4: Difference from iterative retrieval or multi-hop RAG and which component drives the gain**
>
> The key distinction is not retrieval iteration itself, but the explicit separation of two states that standard iterative RAG usually conflates.
> 1. **From the representation side, Content Graph is not standard GraphRAG-style indexing.**  It is a **heterogeneous multimodal evidence graph** over document-native units (paragraphs, tables, figures/captions), and it **jointly evolves** node attributes/topology rather than using a one-shot static index, see example in Fig 3.
> 2. **From the reasoning side, Planning Graph is not standard decomposition.** It is a **maintainable DAG reasoning state**: instead of a linear trace issuing the next query, it tracks dependencies, binds evidence/intermediate answers to nodes, and refines via evidence-sufficiency checks. Progress is thus measured by **closing evidence gaps**, not just taking the next hop.
> 3. **The gain is not due to iteration alone.**
> As shown in **Table 2**, baseline / Content-only / Planning-only / both reach **63.2 / 63.6 / 64.4 / 68.0**, respectively. Thus, the joint gain (**+4.8**) is much larger than either module alone (**+0.4**, **+1.2**). Moreover, **Table 3** shows that DAG refinement further improves **64.8 → 68.0** from **0 → 3** rounds. Together, these results indicate a **non-additive gain** from coupling structured evidence with structured planning.
>
>
> **W2/W4/Q2: Why should the dual-evolving process help, and how does retrieval quality evolve across iterations?**
>
> **The dual-evolving process should help because it improves both the retrieval space and the retrieval policy.**
>
> 1. **It improves the retrieval space.**
>    Content Graph evolution makes retrieval operate over a more coherent **multimodal evidence space** than flat chunks, reducing fragmentation caused by flat chunking. Experiment on representation quality is shown in link https://anonymous.4open.science/r/D2-Reader-8526/Analysis.md
>
> 2. **It improves the retrieval policy.**
>    The Planning Graph turns one-shot retrieval into **gap-driven evidence assembly** through sufficiency checking and refinement, so retrieval is guided by unresolved evidence gaps rather than generic additional rounds.
>
> 3. **The paper already shows that this improves retrieval quality.**
>    In **Fig. 2**, G²-Reader’s recall is consistently above vanilla RAG across tested retrieval budgets; notably, at **k = 5**, G²-Reader reaches **0.863** recall, slightly above vanilla RAG at **k = 20 (0.862)**.
>
> 4. **The gain is not from "more rounds" alone.**
>    If that were the case, a simpler sequential loop should show similar behavior. Instead: The Planning Graph improves recall by **+2.60 points** (**83.70 → 86.30**), whereas the sequential loop changes by only **+0.25** (**79.25 → 79.50**). Most of the gain appears in the first **1–2** refinement rounds.
>
>
> | Iteration | 0 | 1 | 2 | 3 |
> |:---:|:---:|:---:|:---:|:---:|
> | Sequential loop | 79.25% | 79.35% | 79.41% | 79.50% |
> | Plan Graph | **83.70%** | **86.10%** | **86.22%** | **86.30%** |
>
>
>
> **W3/Q3: Robustness to intermediate errors / error propagation**
>
> **The iterative process is reasonably robust to moderate retrieval noise, though not perfect.**
>
> 1. **We directly stress-test error propagation by injecting retrieval corruption.**
>    We replace top-ranked evidence with low-ranked distractors and then run iterative updates.
>
> 2. **Refinement does not simply amplify every intermediate error.**
>    Even under corrupted retrieval, the refinement loop still recovers part of the lost accuracy. For example, with **2 corrupted chunks**, performance improves from **60.50% → 64.41% → 64.92%** over two refinement rounds.
>
> | Corrupted chunks | Iter 0 | Iter 1 | Iter 2 |
> |:---:|:---:|:---:|:---:|
> | 0 | 64.11% | 68.58% | 69.28% |
> | 1 | 61.29% | 62.33% | 63.12% |
> | 2 | 60.50% | 64.41% | 64.92% |
> | 3 | 64.13% | 61.48% | 63.92% |
>
> 3. **The limitation is also clear.**
>    Moderate noise can be partially corrected, but severe corruption can still hurt, which we will discuss explicitly in the revision.
>
> ---
> **Finally:** We really appreciate your insightful suggestions, which have already helped us strengthen both the analysis and presentation of this work. We hope the above clarifications resolve the concerns. If the response is helpful, we would be grateful if you could consider adjusting your score accordingly.

---

> > ### Author Rebuttal · Reviewer_oqR4 · 2026-04-04
> >
> > Thanks for the detailed rebuttal and additional experiments — they are helpful and address my questions.
> > I agree that most of my concerns are now resolved. However, the approach is more like a principled integration and refinement of existing ideas rather than a fundamentally new paradigm.
> >
> > I have increased my score to 4

---

> > > ### Author Response · Authors · 2026-04-07
> > >
> > > Thank you for the thoughtful acknowledgement and for taking the time to review our rebuttal and additional experiments. We are glad that our clarifications addressed your concerns, and we appreciate your recognition of the work and the score update.

---

### Official Review · Reviewer_wLP1 · 2026-03-12

**Soundness:** 3
**Presentation:** 3
**Significance:** 3
**Originality:** 3
**Overall Recommendation:** 4
**Confidence:** 4

**Summary:**

This paper proposes G²-Reader, a dual-graph RAG system for multimodal long-document QA. A Content Graph preserves document structure via iterative VLM-driven node and topology evolution; a Planning Graph decomposes queries into sub-questions via an agentic DAG with execution-verification-replanning loops. On VisDoMBench, G²-Reader with open-source Qwen3-VL-32B achieves 66.21% accuracy, outperforming all RAG baselines and GPT-5 (53.08%).

**Compliance With Llm Reviewing Policy:**

Affirmed.

**Final Justification:**

The rebuttal adequately addressed my original concerns regarding computational cost, evaluation scope, and error propagation through additional experiments and analysis. However, I agree with other reviewers that the novelty is limited, as the contribution lies primarily in the careful integration of well-established components rather than in fundamentally new mechanisms. Given the solid method design, strong empirical results, and thorough experimental coverage, I maintain my score of weak accept.

**Key Questions For Authors:**

1. What is the runtime and API cost of G²-Reader compared to the baselines? Specifically, how many VLM calls are needed for Content Graph construction (per document) and Planning Graph execution (per query)?
2. How does performance change with larger document corpus sizes (e.g., 50 or 500 documents)?
3. The accuracy is evaluated by GPT-4. Have you analyzed how reliable this evaluation is across different subsets or question types?
4. Have you analyzed error propagation in the graph, or explored recovery mechanisms?

**Limitations:**

yes

**Strengths And Weaknesses:**

Strengths:
1. Clean dual-graph design that separates knowledge representation from reasoning control, with each graph addressing a distinct challenge.
2. Open-source model beats GPT-5 by 13+ points, with 121% relative improvement over the base Qwen3-VL-32B.
3. Thorough ablations showing the two graphs provide synergistic gains and good scalability to smaller models.

Weaknesses:
1. The computational cost is not reported. Content Graph evolution requires $T$ VLM calls per node, and Planning Graph requires up to $τ_{max}$ refinement iterations. For a large document, this could be very expensive.
2. The evaluation is limited to VisDoMBench. While it covers 5 subsets, they all follow the same format (5 documents per query, k=5 retrieval budget). How the system performs on different retrieval budgets, document scales, or other benchmarks is unclear.
3. The paper reports GPT-4 based evaluation accuracy, which itself may be noisy. There is no analysis of how evaluation reliability varies across subsets or question types.
4. The Content Graph evolution relies on VLM to update node attributes and topology simultaneously. When the VLM makes errors in early iterations, these could propagate through the graph. The paper does not analyze error propagation or provide recovery mechanisms.

---

> ### Author Rebuttal · Authors · 2026-03-31
>
> We sincerely appreciate your time and thoughtful comments. Below, we respond to each point in detail.
>
> **W1/Q1: Computation time & cost**
>
> 1. **The main extra cost is a one-time offline build, not a recurring per-query burden.**
>    Content Graph construction is an offline preprocessing step reused across subsequent queries / sessions on the same corpus; the recurring deployment cost is only the online Planning/Inference stage.
>
> 2. **The recurring per-query cost remains moderate relative to the gain.**
> Our recurring per-query cost is slightly **lower than GPT-5** (**0.0293  vs. 0.0318 $/query**) while being much more accurate (**68.0% vs. 51.92%**), and far cheaper than **MA-RAG**. A cheaper **rule-based** Content Graph build further reduces the one-time preprocessing cost to **94,223.60 tokens / 26.08s / $0.064**. For one-off QA on a fresh corpus, we agree that the first-use cost is higher, and we will state this trade-off explicitly in the revision.
>
> | Method | Acc. % | Tokens | Time (s) | API cost ($) |
> |:--:|:--:|:--:|:--:|:--:|
> | GPT-5 (single VLM) | 51.92 | 15,291.25 | 131.66 | 0.0318 |
> | Qwen3-VL-32B (single VLM) | 28.39 | 32,261.43 | 102.08 | 0.0163 |
> | MA-RAG | 34.28 | 386,134.13 | 103.37 | 0.19 |
> | **G2Reader** | 68.00 | 54,368.66 | 174.18 | 0.0293 |
>
> **W2/Q2: Evaluation breadth, retrieval budget, and corpus scale**
>
> 1. **Evaluation breadth** The five subsets in VisDoMBench correspond to fundamentally different QA regimes, rather than variations of a single format. We further validate our method on additional benchmark (MMLongBench-Doc) to assess generalization beyond VisDoMBench. The consistent gains across datasets confirm that our improvements stem from the proposed architecture, not from dataset-specific biases.
>
> | Method | MMLongBench-Doc | VisDoMBench |
> |:-:|:-:|:-:|
> | GPT-5 | 38.13 | 53.08 |
> | Qwen3-VL-32B | 29.90 | 29.90 |
> | RAGAnything | 39.1 | 52.25 |
> | VisDoMRAG | 42.11 | 65.01 |
> | **Ours** | **53.5** | **66.21** |
>
> 2. **Retrieval budget sensitivity**
> The table below shows performance across a wide range of budgets (k). We find that **the method is stable across retrieval budgets and corpus sizes**, with only minor variation (~1%). See full table in https://anonymous.4open.science/r/D2-Reader-8526/Analysis.md
>
> |  | k=1 | k=3 | k=5 | k=10 |
> |--|--|--|--|--|
> | ours | 0.6864 | 0.6860 | **0.6989** | 0.6928 |
>
> 3. **Corpus scale robustness**
> We explicitly test larger document pools (up to 50 docs). The table below shows that performance remains largely stable up to 20 documents, and only modestly drops 4% at 50 documents.
>
> | #doc | 5 | 10 | 20 | 50 |
> |--|--|--|--|--|
> | Acc % | 62 | **64** | **64** | 58 |
>
> **W3/Q3: Reliability of GPT-4-based evaluation**
>
> To address this concern, we conducted a human validation study and compared our GPT-4o-based judge against the Word Overlap F1 metric reported in the original VisDoMBench setting. GPT-4o evaluation is nearly identical to human judgment (**64% vs. 62%**) and far more consistent with human evaluation (**0.98 vs. 0.56**) than F1.
>
> | Evaluator | Score | Consistency with human |
> |:-:|:-:|:--:|
> | Word Overlap F1 (rule-based) | 44.03 | 0.56 |
> | GPT-4o judge (Acc) | 64% | **0.98** |
> | Human evaluation (Acc) | 62% | — |
>
> **W4/Q4: Error propagation and recovery.**
>
> We analyze robustness separately for the Content Graph and the Planning Graph. Overall, the iterative process does not show catastrophic error propagation, although severe corruption can still hurt.
>
> 1. **Content Graph errors do not cause catastrophic collapse**
> We perturb the initialized graph at **iter_0** by truncating the summaries of **K** text nodes, then run the standard evolution + QA pipeline. As shown below, performance stays within **58%–68%** rather than collapsing, suggesting that later evolution can partially correct imperfect summaries because each update remains grounded in raw node content and local neighbor context.
>
> |Dataset\K|0|5|10|20|
> |-|-:|-:|-:|-:|
> |Acc%|**62**|**58**|**64**|**68**|
>
> 2. **The Planning Graph is reasonably robust to retrieval noise.**
> We inject retrieval noise by replacing part of the retrieved **top-k** evidence with bottom-ranked items, then run iterative updates. Performance drops only moderately, and refinement still recovers part of the loss; e.g., with **2 injected errors**, accuracy improves from **60.50% → 64.41% → 64.92%**. This suggests that the **DAG reasoning state**, together with **sufficiency checking** and **refinement**, helps absorb moderate retrieval errors rather than directly amplifying them.
>
> |Retrieval noise level|iter=0|iter=1|iter=2|
> |-|-:|-:|-:|
> |Inject 0 error|64.11|68.58|69.28|
> |Inject 1 error|61.29|62.33|63.12|
> |Inject 2 errors|60.50|64.41|64.92|
> |Inject 3 errors|64.13|61.48|63.92|
> ---
> **Finally**: We really appreciate your insightful suggestions.  We hope the above clarifications resolve the concerns. If the response is helpful, we would be grateful if you could consider adjusting your score accordingly.

---

> > ### Author Rebuttal · Reviewer_wLP1 · 2026-04-04
> >
> > Thank you for conducting these additional experiments. Could you clarify what metric is used for "Consistency with human" in W3/Q3's table?

---

> > > ### Author Response · Authors · 2026-04-07
> > >
> > > Thanks for your time and comments, due to the word limits we did not add full explanation on "Consistency with human" in the previous reply.
> > >
> > > Here we refers "Consistency with human" to the **example-level agreement between the evaluator's score and the human score** on the same set.
> > > - For GPT-4o,  is computed as exact agreement on the binary correctness label (1 if both assign the same label, 0 otherwise).
> > > - For Word Overlap F1, which is a continuous metric, we convert it into a binary correctness decision using 0.5 as threshold. We then compute consistency in the same way: 1 if the binarized F1 decision matches the human judgment, and 0 otherwise, averaged over all examples.
> > >
> > > Under this definition, F1 achieves 0.56 agreement (28/50) with human judgment, compared to 0.98 (49/50) for GPT-4o.
> > >
> > > ---
> > > We really appreciate your insightful suggestions.  We hope the above clarifications resolve the concerns. If the response is helpful, we would be grateful if you could consider adjusting your score accordingly.

---

### Official Review · Reviewer_5YoV · 2026-03-13

**Soundness:** 3
**Presentation:** 3
**Significance:** 3
**Originality:** 2
**Overall Recommendation:** 3
**Confidence:** 4

**Summary:**

This paper studies multimodal long-document question answering under the retrieval-augmented generation (RAG) paradigm and proposes G²-Reader, a dual-graph framework designed to improve evidence representation and reasoning over visually rich documents. The system constructs a Content Graph that organizes document elements (paragraphs, tables, figures) and their relationships into a structured multimodal graph, and a Planning Graph that models reasoning as a directed acyclic graph (DAG) of sub-questions.

**Compliance With Llm Reviewing Policy:**

Affirmed.

**Key Questions For Authors:**

1. The paper highlights outperforming GPT-5, but the GPT-5 baseline does not appear to use retrieval or structured indexing. Could the authors include a stronger comparison using GPT-5 within a standard RAG pipeline, or explain why this is not feasible? A stronger baseline here would significantly affect my confidence in the empirical claims.
2. Can the authors provide results on at least one additional multimodal document QA benchmark, or otherwise justify why VisDoMBench alone is sufficient evidence for the broader claims in the paper?
3. The paper compares with and without the Planning Graph, but can the authors compare the DAG-based planner against a simpler sequential retrieval-reasoning loop? This would help determine whether the DAG structure itself is necessary, or whether the gains come mainly from iterative decomposition.
4. Since graph topology and node attributes are updated by a VLM, how stable are the learned/evolved graph structures across runs or prompt variations? If the graph is unstable, that may limit reproducibility; if it is stable, evidence of that would strengthen the paper.

**Limitations:**

I would encourage the authors to discuss limitations more explicitly in two areas:
1. the computational cost and scalability of iterative graph evolution and planning, and
2. potential brittleness of VLM-generated graph relations and evidence sufficiency checks.

**Strengths And Weaknesses:**

Strengths
1. The paper addresses an important problem. Multimodal long-document QA is highly relevant for real-world document understanding, especially when text, tables, and figures must be jointly interpreted.
2. The paper presents a technically coherent system. The decomposition into a document-side representation module and an inference-side planning module is sensible, and the method is described in enough detail to understand the intended pipeline. The empirical section is thorough within the chosen benchmark.

Weaknesses
1. The main weakness is that the conceptual novelty is limited relative to existing work. The Content Graph is close in spirit to graph-based document indexing or GraphRAG-style memory organization, while the Planning Graph resembles existing sub-question decomposition / agentic retrieval frameworks.
2. The evaluation is limited to a single benchmark family. Since the paper makes claims about multimodal long-document QA more broadly, it would be important to see whether the gains transfer to other domain or document types, e.g. MMLongBench.
3. The Content Graph evolution procedure appears potentially expensive because it repeatedly invokes a VLM to update summaries, keywords, and graph neighborhoods. The paper does not provide enough analysis of preprocessing cost, runtime overhead, or scaling behavior with document length.

---

> ### Author Rebuttal · Authors · 2026-03-31
>
> We sincerely appreciate your time and thoughtful comments. Below, we respond to each point in detail.
>
> **W1: The conceptual novelty is limited relative to existing work**
>
> $G^2$-Reader does not simply iterate retrieval over a fixed index; it separates corpus-side evidence representation from query-side reasoning.
> 1. **From the representation side, Content Graph is not standard GraphRAG-style indexing.**  It is a **heterogeneous multimodal evidence graph** over document-native units (paragraphs, tables, figures/captions), and it **jointly evolves** node attributes/topology rather than using a one-shot static index, see example in Fig 3.
> 2. **From the reasoning side, Planning Graph is not standard decomposition.** It is a **maintainable DAG reasoning state**: instead of a linear trace issuing the next query, it tracks dependencies, binds evidence/intermediate answers to nodes, and refines via evidence-sufficiency checks. Progress is thus measured by **closing evidence gaps**, not just taking the next hop.
> 3. **This distinction is empirically substantive.** In Table 2, Planning-only improves 0.632→0.644, Content-only 0.632→0.636, and both reach 0.680; DAG refinement further improves 0.648→0.680 (0→3 rounds). This indicates non-additive gains from coupling structured evidence with structured planning.
>
> **W2/Q2: The evaluation is limited to a single benchmark family.**
>
> The gain transfers beyond VisDoMBench. We additionally evaluate on MMLongBench-Doc (500 samples subset): $G^2$-Reader beats GPT-5 by +15.37 and the strongest baseline by +11.39.
>
> | Method | MMLongBench |
> |:---:|:---:|
> | GPT-5 | 38.13 |
> | Qwen3-VL-32B | 29.9 |
> | RAGAnything | 39.1 |
> | VisDoMRAG | 42.11 |
> | **Ours** | **53.5** |
>
> **W3: The paper does not provide enough analysis of preprocessing cost, runtime overhead, or scaling behavior with document length.**
>
> 1. **The concern partly conflates one-time preprocessing with recurring inference cost.**
> Content Graph evolution is offline; once built, it is fixed and reusable across queries/sessions on the same corpus.
> 2. **The recurring online cost is moderate relative to the accuracy gain.**
> Our full system is slightly cheaper than GPT-5 per query while being substantially more accurate, and much more cost-efficient than MA-RAG.
>
> | Method | Acc. | Time(s) | API-cost |
> |:---:|:---:|:---:|:---:|
> | GPT-5 | 53.08 | 131.66 | 0.0318 |
> | Qwen3-VL-32B | 29.9 | 102.08 | 0.0063 |
> | MA-RAG | 33.07 | 103.37 | 0.1900 |
> | Ours | 66.21 | 174.18 | 0.0293 |
>
> 3. **Corpus scale robustness** We explicitly test larger document pools (up to 50 docs).
> Performance remains largely stable up to 20 documents, and only modestly drops 4% at 50 documents.
>
> | #doc | 5 | 10 | 20 | 50 |
> |---|---|---|---|---|
> | Acc % | 62 | 64 | 64 | 58 |
>
>
> **Q1: Include a stronger comparison using GPT-5 within a standard RAG pipeline.**
>
> In the current version, GPT-5 is intended as a single-VLM reference point. The table below shows that stronger baseline still remains below $G^2$-Reader.
>
> | Method | Acc % |
> |:---:|:---:|
> | GPT-5 (single-VLM) | 53.08 |
> | GPT-5 + standard RAG | 60.13 |
> | Ours | 66.21 |
>
> **Q3: compare the DAG-based planner against a simpler sequential retrieval-reasoning loop**
>
> The table below compares against a simpler sequential iterative retrieval-reasoning loop. A simpler sequential loop reaches 64.65, only +1.05 above no planner (63.6), whereas the DAG planner reaches 68.0, a further +3.35 over the sequential loop.
>
> | Planner | Acc % |
> |:---:|:---:|
> | No planner | 63.6 |
> | Sequential loop | 64.65 |
> | DAG planner | 68 |
>
> **Q4: how stable are the learned/evolved graph structures across runs or prompt variations**
>
> 1. **The evolved Content Graph is reasonably stable across repeated stochastic builds.** To directly test reproducibility, we rebuilt the full Content Graph 3 times per query under T = 0.7 on a stratified sample from all five VisDoMBench subsets; current results are based on 38/50 completed queries, yields Tag Jaccard 0.726, degree-distribution JS 0.032,
>
> | Metric | Mean |
> |:---:|:---:|
> | Keyword Jaccard$\uparrow$ | 0.453 |
> | Tag Jaccard$\uparrow$ | 0.726 |
> | Degree-distribution JS divergence$\downarrow$ | 0.032 |
>
> 2. **The global topology remains highly consistent across runs.** These results indicate that while some local lexical variation exists, the global semantic/structural properties of the graph remain highly stable. We will complete the full 50-query analysis and include the final results in the revision.
>
> | Structural statistic | Mean CV (± std) |
> |:---:|:---:|
> | Node count$\downarrow$ | 0.015 ± 0.080 |
> | Edge count$\downarrow$ | 0.038 ± 0.102 |
> | Mean degree$\downarrow$ | 0.029 ± 0.046 |
> | Global clustering coefficient$\downarrow$ | 0.037 ± 0.033 |
>
> ---
> **Finally:** We really appreciate your insightful suggestions.  We hope the above clarifications resolve the concerns. If the response is helpful, we would be grateful if you could consider adjusting your score accordingly.

---

> > ### Author Rebuttal · Reviewer_5YoV · 2026-04-03
> >
> > Thank you for the detailed rebuttal. The additional experiments and clarifications are helpful, but partial concerns remain only partially addressed. The claimed conceptual novelty is still not fully convincing. The distinctions from existing GraphRAG-style indexing and decomposition-based methods appear incremental rather than fundamentally new. The proposed heterogeneous multimodal evidence graph remains conceptually similar to prior graph-based retrieval frameworks; the primary difference lies in extending the data to multimodal, rather than introducing a fundamentally new modeling paradigm or theoretical insight.

---

> > > ### Author Response · Authors · 2026-04-07
> > >
> > > Thank you for the comment.
> > >
> > > We acknowledge that our method, like other graph-based approaches, uses an explicit graph formulation. However, this should not be conflated with GraphRAG-style indexing, since our Content Graph differs fundamentally in graph ontology and evolution mechanism:
> > >
> > > | Method | Main object | Nodes | Edges | Evolution mechanism |
> > > | --- | --- | --- | --- | --- |
> > > | **GraphRAG** | corpus-level entity graph index | entities | entity relations | offline incremental index update |
> > > | **G²-Reader (Content Graph)** | document-grounded evidence graph | paragraphs + multimodal units | document-grounded semantic dependencies | joint topology + attribute evolution |
> > >
> > > 1. **Different graph ontology.**
> > >
> > >     GraphRAG is entity-centric. Our Content Graph is evidence-centric, with nodes as paragraphs and multimodal units, and edges as semantic and cross-modal evidence dependencies.
> > >
> > > 2. **Different evolution mechanism.**
> > >
> > >     GraphRAG incrementally updates a corpus-level index, while our Content Graph jointly refines topology and node attributes to better represent document evidence structure.
> > >
> > > 3. **Moreover, Content graph evolution is only one component of our method.**
> > >
> > >     We further introduce a **Planning Graph** for query-time reasoning, which explicitly models sub-question dependencies and dynamic refinement. Therefore, our contribution is not simply a multimodal extension of GraphRAG, but a dual-graph framework separating evidence representation from reasoning control.
> > >
> > > ---
> > > We hope the above clarifications resolve the concerns. If the response is helpful, we would be grateful if you could consider adjusting your score accordingly.

---

### Decision · Program_Chairs · 2026-04-30

**Decision:**

Accept (regular)

**Comment:**

After reviewing the original reviews and the authors’ rebuttal, I find that the authors have provided a thorough and responsive engagement with the concerns raised by all reviewers. They addressed the main criticisms—limited benchmark evaluation, computational cost, novelty, and stability of graph evolution—by supplying additional experiments on MMLongBench-Doc, cost and runtime analyses, stability metrics across repeated runs, comparisons with sequential retrieval baselines, and a clear error taxonomy. While several reviewers noted that the conceptual novelty is incremental rather than foundational, the rebuttal successfully demonstrated that the dual-graph design yields non-additive gains beyond standard iterative or GraphRAG-style methods, and the empirical results are solid. The authors also clarified the distinction between offline preprocessing and online inference costs, and provided a lower-cost variant. Given the quality of the rebuttal and the resulting resolution of most concerns, I recommend Accept.